# Relationship Between RAP and Multi-Modal Cerebral Physiological Dynamics in Moderate/Severe Acute Traumatic Neural Injury: A CAHR-TBI Multivariate Analysis

**DOI:** 10.3390/bioengineering12091006

**Published:** 2025-09-22

**Authors:** Abrar Islam, Kevin Y. Stein, Donald Griesdale, Mypinder Sekhon, Rahul Raj, Francis Bernard, Clare Gallagher, Eric P. Thelin, Francois Mathieu, Andreas Kramer, Marcel Aries, Logan Froese, Frederick A. Zeiler

**Affiliations:** 1Department of Biomedical Engineering, Price Faculty of Engineering, University of Manitoba, Winnipeg, MB R3T 5V6, Canada; islama9@myumanitoba.ca (A.I.); steink34@myumanitoba.ca (K.Y.S.); frederick.zeiler@umanitoba.ca (F.A.Z.); 2Department of Anesthesiology, Pharmacology, and Therapeutics, University of British Columbia, Vancouver, BC V6T 1Z3, Canada; donald.griesdale@vch.ca; 3Division of Critical Care, Department of Medicine, University of British Columbia, Vancouver, BC V5Z 1M9, Canada; mypindersekhon@gmail.com; 4Department of Neurosurgery, University of Helsinki and Helsinki University Hospital, 00029 Helsinki, Finland; rahul.raj@hus.fi; 5Section of Critical Care, Department of Medicine, University of Montreal, Montreal, QC H3T 1J4, Canada; bernard.francis@gmail.com; 6Section of Neurosurgery, University of Calgary, Calgary, AB T2N 1N4, Canada; galclare@gmail.com; 7Department of Clinical Neurosciences, University of Calgary, Calgary, AB T2N 1N4, Canada; andreas.kramer@albertahealthservices.ca; 8Hotchkiss Brain Institute, University of Calgary, Calgary, AB T2N 1N4, Canada; 9Medical Unit Neurology, Karolinska University Hospital, 171 76 Stockholm, Sweden; eric.thelin@ki.se; 10Department of Clinical Neuroscience, Karolinska Institutet, 171 76 Stockholm, Sweden; 11Section of Neurosurgery, Department of Surgery, University of Toronto, TO M5S 1A1, Canada; francois.mathieu@mail.utoronto.ca; 12Department of Critical Care Medicine, University of Calgary, Calgary, AB T2N 1N4, Canada; 13Department of Intensive Care, Maastricht University Medical Center+, School of Mental Health and Neurosciences, University Maastricht, 6211 LH Maastricht, The Netherlands; marcel.aries@mumc.nl; 14Section of Neurosurgery, Department of Surgery, Rady Faculty of Health Sciences, University of Manitoba, Winnipeg, MB R3T 5V6, Canada; 15Pan Am Clinic Foundation, Winnipeg, MB R3M 3E4, Canada

**Keywords:** traumatic brain injury, cerebral compliance, RAP, cerebral physiology, multimodal monitoring

## Abstract

Background: The cerebral compliance (or compensatory reserve) index, RAP, is a critical yet underutilized physiological marker in the management of moderate-to-severe traumatic brain injury (TBI). While RAP offers promise as a continuous bedside metric, its broader cerebral physiological context remains partly understood. This study aims to characterize the burden of impaired RAP in relation to other key components of cerebral physiology. Methods: Archived data from 379 moderate-to-severe TBI patients were analyzed using descriptive and threshold-based methods across three RAP states (impaired, intact/transitional, and exhausted). Agglomerative hierarchical clustering, principal component analysis, and kernel-based clustering were applied to explore multivariate covariance structures. Then, high-frequency temporal analyses, including vector autoregressive integrated moving average impulse response functions (VARIMA IRF), cross-correlation, and Granger causality, were performed to assess dynamic coupling between RAP and other physiological signals. Results: Impaired and exhausted RAP states were associated with elevated intracranial pressure (*p* = 0.021). Regarding AMP, impaired RAP was associated with elevated levels, while exhausted RAP was associated with reduced pulse amplitude (*p* = 3.94 × 10^−9^). These two RAP states were also associated with compromised autoregulation and diminished perfusion. Clustering analyses consistently grouped RAP with its constituent signals (ICP and AMP), followed by brain oxygenation parameters (brain tissue oxygenation (PbtO_2_) and regional cerebral oxygen saturation (rSO_2_)). Cerebral autoregulation (CA) indices clustered more closely with RAP under impaired autoregulatory states. Temporal analyses revealed that RAP exhibited comparatively stronger responses to ICP and arterial blood pressure (ABP) at 1-min resolution. Moreover, when comparing ICP-derived and near-infrared spectroscopy (NIRS)-derived CA indices, they clustered more closely to RAP, and RAP demonstrated greater sensitivity to changes in these ICP-derived CA indices in high-frequency temporal analyses. These trends remained consistent at lower temporal resolutions as well. Conclusion: RAP relationships with other parameters remain consistent and differ meaningfully across compliance states. Integrating RAP into patient trajectory modelling and developing predictive frameworks based on these findings across different RAP states can map the evolution of cerebral physiology over time. This approach may improve prognostication and guide individualized interventions in TBI management. Therefore, these findings support RAP’s potential as a valuable metric for bedside monitoring and its prospective role in guiding patient trajectory modeling and interventional studies in TBI.

## 1. Introduction

Traumatic brain injury (TBI) remains a leading cause of mortality and long-term disability worldwide, particularly among young adults [1,2]. Within the neurocritical care setting, continuous bedside monitoring of cerebral physiology is a cornerstone of patient management, offering vital insights into intracranial dynamics. Current clinical management strategies largely rely on guideline-driven interventions, in which intracranial pressure (ICP) is frequently used as a surrogate for assessing intracranial compliance [2,3,4,5]. However, the methodologies associated with this approach are susceptible to inaccuracies, as they often depend on subjective visual assessment of pulse waveform morphology at the bedside—an approach that introduces variability due to inter- and intra-observer inconsistencies. Therefore, the accurate and objective evaluation of continuous compliance metrics is of paramount importance.

One such physiological marker, the cerebral compliance or compensatory reserve index (RAP), has emerged as a promising indicator of cerebral compensatory reserve (and therefore compliance). RAP is derived as the moving Pearson correlation coefficient between ICP and its fundamental pulse amplitude (AMP) [6,7,8,9,10]. Here, the fundamental component refers to the first dominant frequency peak of ICP pulse in the frequency domain. The RAP index ranges from −1 to +1 [6,7,11]. For RAP, as ICP rises, an intact cerebrovascular reactivity mechanism prevents significant changes in AMP; therefore, RAP remains close to 0 [12]. However, when ICP continues to increase and cerebrovascular reactivity becomes impaired, AMP also begins to rise (correlating to ICP) leading to higher positive RAP values, which indicate compromised compliance [12]. If ICP keeps increasing and exceeds a critical threshold, cerebral vessels reach a maximum vasodilation state and cerebrovascular function breaks down, leading to reduced transmission of pulse pressure, i.e., reduced AMP. This leads to lower RAP values (less correlated AMP and ICP) and is characteristic of exhausted cerebral compliance [11,12,13,14].

Recent findings in the hydrocephalus literature have demonstrated the utility of the RAP index in predicting shunt failure, where increasingly positive RAP values have been associated with deteriorating cerebral compliance [6,7,8,9]. Additionally, RAP can be continuously derived at the bedside in patients undergoing invasive ICP monitoring, making it particularly suitable for integration into TBI management protocols.

Despite its potential, the clinical adoption of RAP in TBI cases has been limited, largely due to an incomplete understanding of its characteristics [12]. Previous investigations have focused on characterizing the temporal dynamics of RAP and its constituent signals, ICP and AMP, as well as addressing artifact management strategies [15]. However, there remains a notable gap in the literature regarding the broader cerebral physiologic context and the injury burden of impaired RAP in relation to other continuous multi-modal monitoring (MMM) parameters [15]. A recent systematic review highlighted that various MMM physiological indices consistently exhibited distinct patterns across different states of compliance (i.e., intact, impaired, exhausted) in TBI patients, emphasizing the potential for a more granular analysis of how RAP aligns with these physiologic parameters [11]. However, none of the prior studies had RAP as their primary focus. Furthermore, some of the studies had a small dataset or excluded too many subjects and, therefore, their results were not significant enough to apply the findings clinically. Additionally, a few of the studies showed inconsistent findings that contrasted with most of the existing literature.

Therefore, this study aims to characterize the insult burden of impaired cerebral compensatory reserve in moderate/severe TBI patients in relation to other critical aspects of cerebral physiology by A. describing general associations between RAP and other physiologic signals, B. exploring multivariate covariance patterns using clustering techniques, and C. assessing high-resolution RAP responses to dynamic changes in related variables. Impaired RAP (i.e., positive RAP values) is expected to be associated with increased ICP and AMP, with cerebral autoregulation (CA) measurements expected to be more positive in such states. Additionally, worsening RAP values (i.e., more positive values) are anticipated to correspond with decreased cerebral perfusion pressure (CPP), mean arterial pressure (MAP), regional cerebral oxygen saturation (rSO_2_), and brain tissue oxygenation (PbtO_2_). While these relationships are likely to be more apparent in higher-resolution data, similar trends should still be evident in lower-resolution datasets. This is because, even when high-frequency fluctuations are averaged out in lower-resolution data, the underlying patterns, such as shifts in physiologies associated with impaired compliance, remain preserved. The broader objective is to understand the overall burden of impaired RAP in the context of cerebral physiology, thereby laying the groundwork for its future use in bedside monitoring, patient trajectory modeling, and interventional clinical studies.

## 2. Materials and Methods

### 2.1. Study Design

This retrospective investigation utilized archived human subject data from the Canadian High-Resolution Traumatic Brain Injury (CAHR-TBI) Research Collaborative [16]. At each participating center, high-frequency physiological data were prospectively collected from patients aged 18 years and older who were admitted to the intensive care unit (ICU) with moderate-to-severe TBI. These data were subsequently accessed retrospectively and aggregated to form the CAHR-TBI dataset [16]. For the data collection, each patient was given a anonymized number by the collecting institution (with this patient information being stored only locally in double secured format, as per institutional requirements). The anonymized patient data are then sent to a single site (University of Manitoba) for storage and harmonization. For more information, we refer the interested reader to the following manuscript [16]. The periods of data collection varied by site: Foothills Medical Centre, University of Calgary (2011–2021); Health Sciences Centre Winnipeg (Shared Health Manitoba), University of Manitoba (2019–2023); Maastricht University Medical Center, University of Maastricht (2017–2022); and Vancouver General Hospital, University of British Columbia (2014–2019) [16].

As with the previous studies from our lab group [17,18], patients were included in the database if they met all of the following criteria: they were 18 years of age or older, had a diagnosis of moderate-to-severe TBI with a Glasgow Coma Scale (GCS) score of less than 13, were admitted to the ICU of a participating hospital, underwent both invasive ICP and arterial blood pressure (ABP) monitoring, and had data collection initiated within 24 h of clinical presentation [17,18].

In addition to high-resolution physiological data, the dataset included demographic details (such as age, sex), admission-related characteristics (including admission GCS score and pupillary reactivity), and imaging findings (e.g., Marshall computerized tomography (CT) classification). All data were collected in a fully de-identified manner.

### 2.2. Ethics

Ethical approval for all aspects of data collection and anonymized data sharing across participating centers was obtained from the respective institutional research ethics boards: University of Manitoba Health Research Ethics Board (HREB; protocols H2017:181 (24 May 2017), H2017:188 (24 May 2017), H2020:118 (9 March 2020), H2024:266 (5 September 2024)), University of Calgary Conjoint Health Research Ethics Board (CHREB; protocol REB20-0482 (20 April 2020)), University of British Columbia Clinical Research Ethics Board (CREB; protocol H20-03759 (11 December 2020)), and the Medical Ethics Committee of Maastricht University (protocol 16-4-243 (TA 31 August 2022)). Given that all data were fully anonymized and retrospectively accessed, approval was granted to conduct the study under a waived consent model.

### 2.3. Physiologic Data Acquisition

Consistent with our prior research [15,17,18], high-frequency full-waveform physiological data were recorded from ICU bedside monitors using the Intensive Care Monitoring “Plus” (ICM+) software (version 8.5, Cambridge Enterprise Ltd., Cambridge, UK; http://icmplus.neurosurg.cam.ac.uk, accessed on 5 May 2024), with analog-to-digital converters (Data Translations, DT9804 or DT9826) employed as needed.

For example, in the Manitoba ICUs, ICP and ABP signals were sampled at 100 Hz from analog outputs. ICP was monitored using an intraparenchymal pressure sensor probe (Codman ICP MicroSensor, Codman & Shurtleff Inc., Raynham, MA, USA; NEUROVENT-TEMP, RAUMEDIC, Helmbrechts, Germany; or Camino ICP Monitor, Natus, Middleton, WI, USA) inserted into the frontal lobe, or via an external ventricular drain (EVD; Medtronic, Minneapolis, MN, USA) placed in the lateral ventricle. ABP was measured through a pressure transducer continuously (Edwards, Irvine, CA, USA; Baxter Healthcare Corp. CardioVascular Group, Irvine, CA, USA), which was zeroed at the level of the tragus using either a radial or femoral arterial line [19,20]. Where available, PbtO_2_ was measured using the Licox Brain Tissue Oxygen Monitoring System (Integra LifeSciences Corp., Plainsboro, NJ, USA). The initiation of PbtO_2_ monitoring was determined by the treating clinical team and, as such, was not uniformly applied across all patients or centers [17,18].

rSO_2_ was assessed through near-infrared spectroscopy (NIRS) oximetry, targeting the left (rSO_2__L) and right (rSO_2__R) frontal lobes using the INVOS 5100C or 7100 systems (Covidien-Medtronic, Minneapolis, MN, USA) [17,18]. This measurement was performed in a subset of patients where viable rSO_2_ signal channels were available. Viable channels were defined as those unaffected by underlying lesions, hematomas, or scalp abnormalities. Channels were validated against lesion annotations to confirm that only NIRS signal streams from unaffected regions were included in the analysis [17,18].

The median data monitoring duration was 5811.23 s (IQR: 2872.90 s—9951.18 s)

### 2.4. Signal Processing

Post-acquisition signal processing was performed using ICM+ software, following previously established and published methodologies. Initial steps involved calculating 10-s moving averages of ICP and ABP, updated every 10 s, to minimize data redundancy [15,17,18]. Expert reviewers in neurophysiology and cerebral physiologic signal analysis then manually identified and removed artifacts from non-clean ICP and ABP data. From the cleaned ABP signal, MAP was derived using 10 s moving averages updated every 10 s. Subsequently, CPP was calculated using the formula CPP = MAP − ICP [15,17,18]. AMP was obtained through the Fourier analysis of the fundamental harmonic of the ICP waveform [12,21]. RAP and CA indices were then calculated as moving Pearson correlation coefficients over 30 consecutive 10-s mean windows (i.e., a 5-min analysis window), updated every minute [22,23,24]. Specifically, RAP was derived from ICP and AMP, the pressure reactivity index (PRx) from ICP and MAP, the pulse amplitude index (PAx) from AMP and MAP, RAC from AMP and CPP, and the cerebral oxygenation indices (COx and COx_a) from rSO_2_ and CPP, and rSO_2_ and ABP, respectively [17,18,21,22,25].

The primary derived data was at 1-min intervals. To enable a comprehensive analysis and assess the effects of reduced temporal resolution, the dataset was subsequently down sampled using non-overlapping time windows. This was accomplished with the *resample* function from the *Pandas* library [26]. The original data, calculated at a 1-min resolution, was down-sampled to 5-min, 10-min, 30-min, and 1-h intervals.

### 2.5. Data Cleaning

Several missing data points were present across the dataset. These were excluded from the analysis [15,18,25]. Although artifacts were removed by experts prior to the calculation of derived parameters, some unrealistic values persisted. To ensure data quality, further filtering was applied for ICP- and ABP-derived parameters. Data points were excluded if any of the following criteria were met: ICP > 100 mmHg or <0 mmHg, MAP > 200 mmHg or <0 mmHg, or AMP > 30 mmHg [15,18,25]. For NIRS-derived parameters, visual inspection of scatterplots revealed a cluster of data points at rSO_2_ values below 20, particularly at or near 0, indicating likely recording errors. As a result, a threshold of 20 was established for rSO_2_, and all data points below this value were excluded from the NIRS-derived parameters-related analyses. The entire analysis was conducted using custom Python (version 3.7.16) scripts.

### 2.6. Analysis of General Descriptive Relationships

Based on the results of our previous systematic review, patients were categorized into three groups based on RAP threshold ranges: [−1, 0], [0.4, 1], (1, 0.4) [11]. These threshold ranges were established based on cerebral physiological states (i.e., exhausted, intact, and impaired states of cerebral compliance) and transitions inferred from RAP values [11,15]. Afterwards, general descriptive relationships between RAP and other physiology were explored across the three RAP ranges. Initially, various plots were created for visualization, including scatterplots with piecewise linear regression and boxplots across the entire population. Subsequently, the medians of the cerebral physiological parameters were assessed to explore how RAP values affect other physiological measures. Given the non-parametric nature of the data and the presence of more than two groups, the Kruskal–Wallis test was selected to perform a formal comparison among the groups [17,27]. Finally, the percentage time spent in different ranges of other parameters within different RAP states was observed. For formal comparisons and percentage time calculations, the median value across each patient’s entire recording was used.

Next, a reverse analysis was carried out using the thresholds of other physiological parameters, examining how RAP changed in response. Similarly, these changes were analyzed using the median measurements and statistical comparisons. Furthermore, the percentage time RAP spent within each of the RAP states at these threshold ranges was examined. The thresholds for various parameters were as follows: ICP (20 mmHg, 22 mmHg) [2,23], CPP (60 mmHg, 70 mmHg) [2], rSO_2_ (60, 70, 80, 90) [28,29], PbtO_2_ (15 mmHg, 20 mmHg) [20,30,31], PRx (0, +0.25, +0.35), PAx (0, +0.25), RAC (0) [23,32], COx/COx-a (0, +0.20) [29]. “The thresholds used for physiological parameters, including ICP, CPP, CA indices (PRx, PAx, RAC, COx/COx-a), rSO_2_, and PbtO_2_, were based on prior published studies and established clinical guidelines.” These values are not arbitrary but represent empirically validated cutoffs linked to clinically relevant outcomes, such as impaired cerebrovascular reactivity, secondary injury progression, and mortality in TBI patients.

Mann–Whitney U-test was applied for pairwise comparisons between groups, due to the non-parametric nature of the data [15,17,18].

### 2.7. Application of Algorithms for Parameter Clustering

To quantify covariance patterns in multivariate space for the previously discussed physiological parameters, clustering methods, namely agglomerative hierarchical clustering (AHC), principal component analysis (PCA), and K-means clustering (KMCA), were utilized [25,29,33].

AHC is an ML algorithm used for hierarchical clustering of data. The method works by initially treating each data point as an individual cluster, and then iteratively merging the closest clusters based on a defined distance metric. This process continues until all data points are clustered into a single group [34,35]. In this study, the Euclidean distance metric was employed, and dendrograms were generated for visualization of the output. By examining the clustering and merging distances in the dendrogram, it became possible to identify patterns of similar behavior among parameters across various instances, as well as to uncover inherent subgroups within the data that displayed comparable dynamics. This analysis was conducted using the *hierarchy* submodule from the *SciPy* library, with the adequacy of fit of the resulting dendrogram quantified using cophenetic correlation coefficients [36].

Next, PCA was employed, which is a statistical technique that reduces the dimensionality of multivariate data while retaining as much variance as possible. PCA transforms the original set of variables into a new set of orthogonal components, known as principal components, ranked by the amount of variance they capture [37,38]. In this study, a PCA biplot was constructed using the first two principal components (PC1 and PC2), with the directional vectors of the physiological parameters projected onto this reduced space. Parameters that exhibited alignment in a similar direction within the biplot were interpreted as being positively correlated, indicating that they share similar variance structures and contribute in a comparable manner to the principal components. Furthermore, to assess the contribution and effectiveness of the principal components, scree plots of the explained variance ratio and cumulative explained variance were analyzed. This analysis was performed using the *PCA* class from the *scikit-learn* library [39].

Lastly, KMCA is a widely used semi-supervised learning algorithm that partitions data into k distinct clusters by iteratively assigning points to the nearest centroid and updating centroids based on a similarity measure [40,41]. In this analysis, the Euclidean distance was used as the similarity measure, and the elbow method was applied to identify the optimal number of clusters (k). This technique involved plotting the within-cluster sum of squared errors (WCSS) against varying values of k. As k increased, WCSS typically decreased; however, after a certain point, the rate of decrease sharply declined. The “elbow” point on the plot, where the curve starts to level off, indicated the optimal number of clusters. The *KMeans* class from the *scikit-learn* library was used to carry out this analysis [42].

These three methods were chosen as they allow for both dimensionality reduction and identification of subgroups within the data, thereby enabling a more robust characterization of shared variance structures and covariance patterns across parameters.

To execute these algorithms, all parameters require complete data, thus necessitating the exclusion of patients with any missing values. However, as seen in Section 3.1, out of 379 patients, only 133 and 116 patients had available data for both NIRS-derived parameters (rSO_2__L, rSO_2__R, COx_L, COx_R, COx-a_L, COx-a_R) and PbtO_2_, respectively. Therefore, using all parameters would have resulted in insufficient data, reducing analysis efficiency. To overcome this, the population was divided into three groups: (a) ICP-ABP-derived parameters (ICP, AMP, MAP, CPP, PRx, PAx, RAC), (b) ICP-ABP-derived parameters with NIRS-derived parameters, and (c) ICP-ABP-derived parameters with PbtO_2_, and separate analyses were conducted on these groups to ensure a more comprehensive and effective approach.

The analysis was initially conducted on the original data (i.e., 1-min). Later, the entire analysis was repeated on lower-resolution datasets. Afterwards, the whole dataset was divided into three RAP states (intact, impaired, severely impaired), as defined in our previous systematic review [11], and the analysis was conducted on these sub-groups across all temporal resolutions.

### 2.8. High-Frequency Temporal Analysis of RAP vs. MMM Relationships

In regard to our research objective, high-frequency temporal analysis of RAP vs. MMM relationships would enable us to capture the dynamic, time-resolved interactions between RAP and other parameters. This section of the analysis includes vector autoregressive integrated moving average (VARIMA), impulse response function (IRF) plots with corresponding quantitative evaluations, Granger causality testing, and cross-correlation analysis.

The analysis was conducted on the 1-min, 5-min, and 10-min resolution data. Lower-resolution data were excluded as these methods are unlikely to capture any meaningful dynamics at such coarse temporal resolutions. This is because at lower resolutions, short-term variability is smoothed, resulting in too few data points for robust analysis, thus reducing the sensitivity of these methods.

The analysis was also further extended to the sub-group level. For Granger causality testing and cross-correlation analysis, each patient’s recording was divided into three segments based on the previously defined RAP thresholds, with each segment analyzed separately. However, this segmentation approach was not feasible for IRF analysis involving VARIMA models, as the RAP < 0 and 0 ≤ RAP ≤ 0.4 segments often lacked sufficient data to reliably fit the optimal VARIMA models for many patients. To address this, the median RAP value of the entire recording period was calculated for each patient, and patients were then grouped into the three RAP states based on these median values.

#### 2.8.1. Application of VARIMA IRF

VARIMA is a multivariate time series modeling approach that extends the autoregressive integrated moving average (ARIMA) framework to capture temporal dependencies and interactions across multiple variables [29,33,43]. It accounts for both autoregressive (AR) and moving average (MA) components, along with differencing (I) to ensure stationarity. Impulse Response Function (IRF) analysis was then applied to the fitted VARIMA models [29,33] to quantify the effect of a one-time impulse in a given physiological variable (e.g., ICP, MAP) on RAP over subsequent time points, allowing for the temporal propagation of influence to be assessed.

To determine the optimal AR order (i.e., p-order) and MA order (i.e., q-order) of the VARIMA model, the optimal ARIMA orders were first determined. As ARIMA modeling requires stationarity, and prior analyses confirmed that the dataset was non-stationary [15,17,18], first-order differencing was applied to achieve stationarity. Optimal ARIMA models were then selected for each patient using the Akaike information criterion (AIC), which was preferred over the Bayesian information criterion (BIC) and log-likelihood (LL) due to its balanced trade-off between model fit and complexity [15,17,33]. Since the objective was to assess the influence of other physiological parameters on RAP, pairwise VARIMA models were computed for each RAP-X combination, where X denotes a secondary physiological variable. The VARIMA p-order for each RAP-X pair was determined by multiplying the respective ARIMA p-orders of RAP and X [25,33,44]. The VARIMA q-order was then computed by summing the individual ARIMA q-orders [33,44], following approaches established in previous work. All VARIMA models were fitted individually for each patient.

While determining the optimal orders for the VARIMA model, p- and q-orders were capped at 10. This constraint was applied to prevent overfitting and to preserve the practical applicability of the model for real-time monitoring. Moreover, prior studies have demonstrated that limiting the model orders to 10 did not change the results significantly [25].

Afterwards, the IRF coefficients were derived from the fitted VARIMA model for each RAP–X pair at the individual patient level. These coefficients were then used to generate IRF plots, offering a graphical depiction of the effect of a one-unit orthogonal impulse of first-order differenced X (∆X) on first-order differenced RAP (∆RAP) over time. To estimate the population-level confidence intervals, a bootstrapping approach was employed [33,45]. This involved repeatedly sampling subsets of the data and recalculating the IRF, with a standard percentile-based bootstrap interval generated from a number of iterations to quantify variability and uncertainty in the response estimates [33,45]. In this study, 100 iterations were used.

Given the high variability across IRF plots and the limitations of relying solely on visual interpretation in large datasets, a simplified classification approach was adopted to distinguish between “more responsive” and “less responsive” models [33]. This involved normalizing each impulse response to its original variable. A response was considered “more responsive” if its absolute value exceeded 0.001 (i.e., a ≥0.1% change in the normalized scale) within steps 11 to 15, consistent with thresholds used in prior studies [25,33]. The observation in these steps was effective since immediate responses might reflect noise or autoregulatory transients, while sustained changes after a short delay are more likely to reflect true, biologically meaningful interactions. Additionally, observing steps after 15 could reduce the practical utility of the analysis for real-time monitoring. To be noted, each step represents one unit of the model’s temporal resolution.

The *VARMAX* class from the *statsmodels* library was employed to fit the VARIMA optimal model [46]. While the *VARMAX* class facilitates the implementation of VARIMA models, it does not include a specific argument for the differencing component (i.e., d-order) [46]. To address this, first-order differenced data was used as input, effectively setting the differencing order (d-order) to 1 during the VARMAX operation. In this manner, the VARIMA model was successfully fitted. The IRF analysis was carried out using the *IRAnalysis* class from the *statsmodels* library [47].

#### 2.8.2. Application of Granger Causality Testing

To assess potential directional relationships between RAP and other physiological parameters, Granger causality testing was employed. This method determines whether past values of one time series contain statistically significant information that helps predict the future values of another series, beyond the information contained in its own past values [48,49]. The test was conducted at the patient level for each ∆X→∆RAP pair, where associated *p*-values and F-statistics were calculated and compared against those from the corresponding ∆RAP→∆X direction. Initially, *p*-values were assessed for each ∆RAP–∆X pair. If only one direction yielded a statistically significant *p*-value (*p* < 0.05), that direction was considered to represent Granger causality for the pair. In cases where both directions were significant, the direction with the higher F-statistic was selected as the causal direction. If neither *p*-value was significant, the observation was classified as having no Granger causal directionality. First-order differenced data were used as inputs to ensure stationarity, a prerequisite for performing Granger causality testing [25,33].

This analysis was also applied across the discussed resolutions and sub-groups. The *grangercausalitytests* class from the *statsmodels* library was used to conduct the test [50].

#### 2.8.3. Application of Cross-Correlation

Cross-correlation is a statistical method used to measure the similarity or degree of association between two time series as a function of the lag of one relative to the other. In physiological data analysis, it helps to identify whether changes in one signal lead, lag, or coincide with changes in another [51,52].

To perform this analysis, each variable was first mean-centred to ensure a zero mean, preventing biased cross-correlation results that could otherwise occur due to offset signals. The resulting cross-correlation values were then normalized by dividing by the product of the standard deviations of the mean-centred signals and the signal length. This standardization confined the correlation values within the range of −1 to 1, comparable to Pearson’s correlation, and eliminated the influence of differing signal scales between RAP and the corresponding variable. Finally, the entire analysis was performed on first-order differenced data to prevent misleading correlation results and to more accurately capture the true lagged relationships between the variables. The lag position of the peak correlation provides insight into the temporal relationship between the signals: a peak near lag 0 suggests synchronous changes, a positive lag indicates that the other variable precedes changes in RAP, and a negative lag indicates RAP precedes changes in the other signal. Additionally, the magnitude of the peak correlation reflects the strength of the association.

This analysis was performed at the population level across three different data resolutions. For each patient, the maximum cross-correlation value and the corresponding absolute lag value at which this maximum occurred were recorded. Median values of these metrics were then calculated across all patients to provide a summarized overview of the results. Subsequently, this was conducted at the sub-group level across all the resolutions. The *correlate* class from the *scipy* library was used to conduct cross-correlation [53].

## 3. Results

### 3.1. Patient Demographics

A total of 379 patients were included in the study. Of these, 295 were male. The median age was 38 years, with an interquartile range (IQR) of 24 to 55 years. Additional key demographic and clinical characteristics of the patient cohort are summarized in Table 1. This table also presents the number of patients with NIRS-derived parameters and PbtO_2_, and the median percentage of each RAP state at the patient level.

### 3.2. Graphical Relationships Between RAP and MMM Cerebral Physiologic Variables

After dividing the data according to three RAP threshold ranges ([0.4, 1], (0, 0.4), and [−1, 0]), scatterplots were drawn with the piecewise linear regression for all the parameters, illustrated in Appendix A. For this figure, 1-min resolution was used. The slope of the straight line for each threshold range obtained from the linear regression is illustrated in Table 2.

As presented in the table and figures, ICP and AMP displayed similar slope signs (i.e., positive/negative slopes) across all segments. The significant positive slopes of ICP and AMP in the RAP positive range indicate that both parameters increased as RAP increased, consistent with the theoretical understanding of RAP. However, in the RAP < 0 state, AMP showed a slightly smaller negative slope (−0.24). Regarding MAP and CPP, the reduction in their values in the RAP < 0 state was evident in both figures. Furthermore, CPP decreased with the increase of RAP in its positive range.

Theoretically, as cerebral compliance worsens, CA indices should increase and remain positive, indicating impaired cerebrovascular reactivity. This pattern was observed in the RAP < 0 state for most CA indices (PRx, RAC, COx_R, COx-a_L, COx-a_R), where regression lines were in the positive region and showed an upward trend as RAP decreased toward −1. However, interestingly, the opposite occurred in the RAP > 0.4 state, where all CA indices decreased toward negative values as RAP increased. Additionally, at the 0 ≤ RAP ≤ 0.4 state, most slopes (RAC, COx_L, COx-a_L, COx-a_R) suggested that reactivity worsened as RAP decreased, with regression lines remaining in the positive region. These findings suggested that impaired reactivity was associated with lower values of RAP. This was due to the transitional phase of cerebral compliance when it worsens from an impaired to an exhausted state.

The NIRS-derived rSO_2__L and rSO_2__R showed a decrement with the increase of RAP towards +1, indicating reduced oxygen saturation. Interestingly, rSO_2__L and rSO_2__R also decreased as RAP moved towards −1 from 0. PbtO_2_, in contrast, was found to be elevated in the RAP > 0.4 state. In the other two states, the change rate was minimal, i.e., the slopes were much smaller (−2.43 and −1.22).

Appendix A presents boxplots for all parameters across the entire population. The patterns observed align with those in the scatterplots, with most parameters exhibiting consistent increases or decreases within specific RAP states. This further validates the findings of the plots.

### 3.3. Comparison of MMM Cerebral Physiology Across RAP Threshold Categories

Afterwards, the median values of different physiological parameters, along with IQRs, were calculated across the three RAP states and presented in Appendix A. This table also includes the *p*-values obtained from the Kruskal–Wallis test conducted across the three RAP states. Furthermore, the percentage of time spent in each RAP state for various parameters was also included in this table. Most of these median values were consistent with the findings illustrated in Appendix A. However, while ICP was expected to have higher median values in the RAP < 0 state and CPP was anticipated to be lower compared to the 0 ≤ RAP ≤ 0.4 state, the results contradicted these expectations. This discrepancy was likely due to uneven data distribution, particularly the limited availability of data in the RAP < 0 state, which might have caused unexpected shifts in median values. Furthermore, consistent with previous figures, the median values of all CA indices indicated that impaired autoregulation (i.e., higher CA index values) was associated with lower RAP values, with the exception of PAx. Regarding PbtO_2_, the 0 ≤ RAP ≤ 0.4 state exhibited the lowest median PbtO_2_ value, while the other two RAP states showed increasing trends, with the RAP > 0.4 state having the highest PbtO_2_ value.

In the formal statistical comparison analysis, ICP, AMP, MAP, PRx, and RAC demonstrated significant differences (*p* < 0.05) across the RAP states, based on the Kruskal–Wallis test. The percent time spent in these RAP states across predefined thresholds of different physiological parameters was also depicted in the table, which showed a similar trend to these findings.

#### Comparison of RAP Behaviour Across Different MMM Critical Thresholds

Next, a reverse analysis was conducted with other physiological thresholds. First, the median values for each parameter across the segments were calculated, along with the corresponding *p*-values obtained from the Mann–Whitney U-test (for two groups) and the Kruskal–Wallis test (for three or more groups), enabling a formal comparison between the groups. These results are presented in Appendix A. Furthermore, this table also contains the % time spent with RAP within the three RAP states.

As shown in the table, while higher RAP was associated with higher ICP and CPP threshold ranges, for the rest of the parameters, higher RAP was associated with lower threshold ranges. In terms of CPP thresholds, the association of higher RAP with higher CPP threshold contradicted the previous observations from the RAP-CPP scatterplots (Appendix A), boxplots (Appendix A), and median values from Appendix A. This discrepancy is likely because the scatterplots and table provided a more continuous representation of the data, capturing localized variations or specific trends within RAP states.

From the analysis of the percentage of time spent across different RAP threshold ranges, it was evident that the RAP < 0 state was associated with a higher proportion of time spent at elevated ICP levels. Specifically, for ICP ≥ 20 and ICP ≥ 22, the percentages were 6.25% and 6.21%, respectively, compared to 4.69% and 4.66% for their corresponding lower thresholds. This suggests that an exhausted RAP state (i.e., RAP < 0) was more frequently associated with elevated ICP values (ICP ≥ 20 and ICP ≥ 22), consistent with the previous findings in this study. A similar trend was observed across most CA indices, where higher CA index values corresponded to greater percentages at higher threshold ranges. In the RAP < 0 state, the percentage for PRx ≥ 0 was 6.12%, whereas it was 3.12% for PRx < 0.

Furthermore, in the formal comparison analysis across the different thresholds of the parameters, RAP showed significant differences for PRx (thresholds 0, +0.25, +0.35), RAC (threshold 0), rSO_2__L (threshold 70), rSO_2__R (thresholds 80, 90), COx_L (threshold 0), as reported in Appendix A.

### 3.4. Clustering of the Parameters

#### 3.4.1. Agglomerative Hierarchical Clustering (AHC)

As mentioned in Section 2.7, all the semi-supervised ML approaches were conducted separately for three segments of data across different resolutions to ensure a more comprehensive analysis. Initially, AHC was applied to the 1-min resolution data, and the resulting dendrograms are presented in Figure 1.

As illustrated in the figure, for the ICP-ABP-derived parameters, ICP-AMP-RAP first clustered with MAP-CPP to form a large group, which then merged with the PAx-RAC-PRx cluster, completing the dendrogram as shown in Figure 1a. When NIRS-derived parameters were included, the ICP-AMP-RAP cluster joined with rSO_2__L-rSO_2__R, along with MAP-CPP, to form a larger cluster. This cluster subsequently merged with the PAx-RAC-PRx group, and finally with the COx_L-COx_R-COx-a_L-COx-a_R cluster to complete the dendrogram, as depicted in Figure 1b. This structure closely resembled the previous dendrogram, with the only notable difference being the addition of rSO_2__L and rSO_2__R to the ICP-AMP-RAP-MAP-CPP cluster. In the last dendrogram shown in Figure 1c, incorporating PbtO_2_ into the ICP-ABP-derived parameters led to RAP clustering with PbtO_2_, which then grouped with MAP and CPP to form a large RAP-PbtO_2_-MAP-CPP cluster. This was later joined by the PAx-RAC-PRx-ICP-AMP cluster to complete the dendrogram.

This analysis was also conducted across different data resolutions, with the resulting dendrograms presented in Appendix A. For both the ICP-ABP-derived parameter analysis and the combined ICP-ABP and NIRS-derived parameter analysis, the dendrograms consistently showed the same clustering patterns as observed in the 1-min resolution (Figure 1). However, in the analysis involving ICP-ABP-derived parameters and PbtO_2_, RAP did not cluster directly with PbtO_2_. Instead, it initially grouped with ICP and AMP, and this cluster later merged with the MAP-CPP-PbtO_2_ group. This pattern was consistently observed across all other lower resolutions. The difference observed at the 1-min resolution compared to the lower resolutions was likely caused by the higher temporal granularity of the data. The cophenetic correlation values for the dendrograms are illustrated below in Table 3:

Across all groups and resolutions, the cophenetic correlation coefficients were consistently high, with most cases nearing 0.9, indicating that the agglomerative hierarchical clustering provided a strong and reliable representation of the data structure.

Furthermore, AHC analysis was performed across different RAP states at all the discussed resolutions. The resulting dendrograms are presented in Appendix A. Across all dendrograms, most of the larger clusters were consistent with the findings from the analysis on the entire population. However, noticeable variations appeared within the smaller clusters. For instance, in several dendrograms, particularly for the RAP < 0 and 0 ≤ RAP ≤ 0.4 states, RAP did not cluster directly with ICP-AMP. Another interesting observation was the frequent direct clustering of PbtO_2_ with RAP, suggesting a potentially strong association between these two. Appendix A presents the cophenetic correlation values from the subgroup analysis, with most values exceeding 0.85, suggesting that the AHC models demonstrated a strong overall fit.

#### 3.4.2. Principal Component Analysis (PCA)

At first, PCA was conducted on 1-min resolution data. To assess the contribution and effectiveness of the principal components, scree plots of the explained variance ratio and cumulative explained variance were analyzed, as depicted in Appendix A. As illustrated in the figures, the first two principal components accounted for 54%, 39%, and 51% of the variance in the data for the respective predefined analysis groups. Notably, the ICP-ABP-derived and NIRS-derived parameter group explained less than 50% of the variance (39%), likely due to the higher number of parameters included in that group. The corresponding biplots for these data groups are shown in Figure 2.

As depicted in Figure 2, for the ICP-ABP-derived parameters, ICP-AMP-RAP, MAP-CPP and PAx-RAC-PRx formed groups aligning in a similar direction (Figure 1a). Incorporating NIRS-derived parameters to this group resulted in forming the ICP-AMP-RAP-rSO_2__L-rSO_2__R group, whereas COx and COx-a parameters formed another group in a different direction, as depicted in Figure 1b. Notably, in both cases, RAP appeared slightly offset from the core direction of its group. Lastly, for the ICP-ABP-derived parameters and PbtO_2_, Figure 1c illustrates the formation of a MAP-CPP-RAP-PbtO_2_ group, while ICP-AMP-PAx-RAC-PRx aligned along a different axis. These patterns were consistent with the cluster structures observed in the AHC dendrograms presented in Figure 1.

Subsequently, this analysis was extended to the lower-resolution data. The scree plots corresponding to these resolutions are presented in Appendix A. The proportion of variance explained by the first two principal components remained comparable to that observed in the 1-min resolution data. The PCA biplots for lower-resolution data are presented in Appendix A. Across these resolutions, the majority of biplots demonstrated grouping patterns consistent with those observed at the 1-min resolution. While some minor deviations were present—for instance, RAP was positioned between MAP and CPP in FAppendix A—the larger clusters remained largely unchanged. The biplots of sub-group analysis are illustrated in Appendix A. This analysis showed certain deviations in the smaller groups from the results observed in the whole population, especially within the RAP < 0 and 0 ≤ RAP ≤ 0.4 states. Notably, in Appendix A, PRx was situated closer to the ICP-AMP-RAP cluster rather than with the PAx-RAC group, as seen in previous results. In contrast, during the RAP > 0 state, RAP consistently aligned more closely with MAP and CPP across all resolutions.

#### 3.4.3. K-Means Clustering Analysis (KMCA)

As outlined in Section 2.7, the elbow method was employed to identify the optimal value of k, with its visual representation provided in Figure 3.

As illustrated in the figures, at k = 1 and k = 2, there were sharp drops in WCSS. Beyond k = 3, the reduction in WCSS became more gradual and consistent, indicating diminishing returns with increasing k. This was noticeable across all the groups. This pattern suggests that k = 3 is the optimal choice for KMCA and can be applied across all data groups. Using the optimal k = 3, KMCA was applied across the groups at the 1-min resolution data, and the three clusters that were found are illustrated in Table 4.

The first row of the table presents the clustering results at the 1-min resolution. The three clusters identified through KMCA showed some differences compared to those observed in the AHC dendrograms and PCA biplots. Notably, in both the ICP-ABP-derived parameters group and the ICP-ABP-derived parameters with PbtO_2_ group, RAP was not clustered with its source signal, ICP. Additionally, across all data groups, RAP consistently clustered with the ICP-ABP-derived CA indices (i.e., PRx, PAx, and RAC).

Additionally, this analysis was extended to lower-resolution data. Using the elbow method in the same manner, the optimal k was determined, with graphical illustrations provided in Appendix A. Based on these visualizations, an optimal k of 3 was consistently selected for all data groups and resolutions. The KMCA clustering results for each resolution were summarized in Table 3, which clearly demonstrates that the cluster structures remained consistent across all resolutions. Next, sub-group analysis was conducted. The remaining figures in Appendix A display the elbow method plots for these sub-groups. In this context, the optimal number of clusters was also determined to be 3 for all RAP states. The KMCA clustering outcomes were summarized in Appendix A, indicating that the cluster structures remained consistent across all RAP states and resolution levels.

### 3.5. High-Frequency Temporal Response Patterns of RAP to MMM Changes

#### 3.5.1. Vector Autoregressive Integrated Moving Average (VARIMA) Impulse Response Function (IRF) Analysis

The optimal VARIMA model was determined for each patient and each RAP-X pair across all analyzed resolutions. From these, the median values of the model orders were computed, yielding the median optimal VARIMA models summarized in Appendix A. Following model fitting, impulse response function (IRF) coefficients were extracted and plotted over time. Appendix A present sample IRF plots for a representative patient across all the discussed RAP-X pairs and resolutions. In most instances, the responses remained within the 95% confidence interval, suggesting statistical reliability. Given the observed variability in responses, a thresholding approach was applied to the normalized IRF data, as outlined in Section 2.8.1, to classify responses as either “more responsive” or “less responsive.” The outcome for the “more responsive” group was summarized in Table 5. In some cases, insufficient data points prevented the test from yielding results; these instances were marked as not applicable (NA) in the table.

The table presents the percentage and count of cases classified as belonging to the “more responsive” group for each parameter pair across resolutions. At the 1-min resolution, ICP-ABP-derived parameters consistently exhibited higher responsiveness percentages compared to NIRS-derived parameters and PbtO_2_. While a general decline in responsiveness percentages was observed at lower resolutions, the trend of ICP-ABP-derived parameters showing higher responsiveness largely persisted. The only notable exception was rSO_2__L and rSO_2__R, which demonstrated comparatively higher responsiveness percentages at the lower resolutions.

As outlined in Section 2.8, subgroup analysis was also performed based on patients’ median RAP values. The corresponding results are presented in Appendix A. As indicated, the number of patients within the RAP < 0 category was limited—only six for ICP-ABP-derived parameters, one for NIRS-derived parameters, and four for PbtO_2_. Similarly, the 0 ≤ RAP ≤ 0.4 category included relatively few patients. Due to these small sample sizes, the findings from these two RAP states may lack sufficient reliability. In contrast, the RAP > 0.4 group included a larger number of patients and produced results largely consistent with those observed in the full cohort analysis.

#### 3.5.2. Granger Causality Testing

By comparing the F-statistic values of ∆X→∆RAP and ∆RAP→∆X for each patient example, the direction of the causality was determined. Since the focus of the research is to check how RAP responds to changes in another variable, the percentage of ∆X→∆RAP was calculated and summarized in Table 6.

As shown in the table, most parameter pairs had a large proportion of non-significant (NS) cases, indicating that, in many instances, there was no statistically significant Granger causality in either direction. Among the significant results, ∆PRx, ∆PAx, and ∆RAC were the only parameters for which ∆X→∆RAP was more frequently significant than the reverse direction (∆RAP→∆X) at the 1-min resolution. However, at lower resolutions, ∆ICP, ∆AMP, ∆MAP, and ∆CPP more often showed significant influence on RAP, suggesting resolution-dependent patterns of directional association. A similar pattern was observed at the subgroup level, as shown in Appendix A. However, given that the majority of patient cases yielded non-significant results, the directional findings from the remaining data should not be interpreted as broadly generalizable.

#### 3.5.3. Cross-Correlation Analysis

Initially, cross-correlation analysis was performed at the patient level across the entire population, with the results presented in Table 7.

Among the variables, ∆ICP, ∆AMP, and the ICP-ABP-derived CA indices—∆PRx, ∆PAx, and ∆RAC—exhibited relatively higher median correlation values with ∆RAP, particularly when compared to NIRS-derived CA indices. These parameters also showed median maximum lags of 0 or 1 (i.e., the lag values at which peak correlation occurred), suggesting they fluctuated synchronously with ∆RAP. The findings were largely consistent across all resolutions. Notably, in the lower-resolution data, ∆MAP and ∆CPP showed correlation values comparable to or greater than those of ∆ICP and ∆AMP, with similarly low median maximum lags, indicating synchronous behavior with ∆RAP.

Appendix A present the results from the subgroup analysis. The RAP > 0.4 subgroup largely resembled the trends observed in the full dataset. In contrast, the RAP < 0 and 0 ≤ RAP ≤ 0.4 subgroups displayed notable differences. These groups showed generally higher median maximum lags across most parameter pairs, although ∆PRx, ∆PAx, and ∆RAC showed the lowest lag values. Interestingly, in these states, the NIRS-derived parameters demonstrated stronger correlation magnitudes than the ICP-ABP-derived indices. Additionally, a higher frequency of NA cases was observed, especially at lower resolutions, highlighting the impact of insufficient data on the analysis.

## 4. Discussion

Our objective was to investigate the burden of impaired cerebral compliance, defined as RAP, in moderate-to-severe TBI patients and how it relates to other key components of MMM cerebral physiology. First, a detailed descriptive analysis to assess statistical associations across different sub-groups and threshold ranges was performed. Next, clustering techniques were applied to examine parameter clustering, with a particular focus on how RAP grouped with other variables. Finally, high-frequency temporal response patterns were analyzed to evaluate how RAP dynamically responds to changes in other MMM parameters. Several key aspects deserve highlighting.

### 4.1. General Descriptive Relationships

Firstly, the analysis of general descriptive relationships showed that higher ICP and AMP were associated with impaired RAP, whereas in the exhausted RAP state, ICP remained high but AMP was comparatively lower. This explains the negative RAP values in the exhausted cerebral compliance state. Furthermore, CPP was observed to be lower during impaired and exhausted RAP states, consistent with the concept of diminished perfusion resulting from compromised cerebral autoregulation [13,54]. Clinically, this suggests that RAP can serve as an integrated marker reflecting the combined impact of elevated ICP, reduced AMP, and diminished CPP, thereby providing clinicians with a more comprehensive indicator of cerebral reserve status rather than considering each parameter in isolation.

In the context of CA indices, theoretical expectations indicate that worsening compliance is associated with positive CA indices [12,13,54]. As such, both impaired and exhausted compliance states should exhibit positive CA indices, with the exhausted state showing the most elevated values [12,13,54]. The conducted analyses aligned with these expectations in the exhausted RAP state but not in the RAP > 0.4 and 0 ≤ RAP ≤ 0.4 ranges. This discrepancy can be explained by the typical progression of TBI patients into an exhausted compliance state. Initially, in an impaired compliance phase, patients may experience further deterioration, leading to arteriolar dilation failure and discontinuity in cerebral blood flow (CBF). This reduces the transmission of arterial pulse pressure to the intracranial compartment, resulting in decreased AMP. Consequently, RAP begins to decline and may pass through the 0 ≤ RAP ≤ 0.4 range, which is normally indicative of intact compliance. However, in this context, it represents a transitional phase toward negative RAP values, indicating exhausted compliance. Therefore, the analyses showed that CA indices were higher when RAP was near zero or negative, compared to when RAP was greater than 0.4. This transitional behavior also highlights the importance of not interpreting RAP thresholds in isolation but rather in the broader temporal and physiological trajectory of the patient, as misclassification may lead to delayed recognition of compliance exhaustion.

This relationship between cerebral compliance and cerebral autoregulation is also supported by pre-clinical and clinical literature examining the pathophysiological basis of ICP–AMP dynamics and their link to autoregulatory function [55,56,57]. Experimental studies from the Cambridge group and subsequent animal model investigations in rabbits have demonstrated that as ICP rises toward critical thresholds, a distinct break-point in AMP occurs [57]. This inflection point has been theorized to represent the critical closing pressure of small- to medium-sized cerebral vessels [55,56]. Beyond this point, progressive ICP elevations lead to diminished transmission of arterial pulsatility into the intracranial compartment, reflected by falling AMP values and corresponding negative RAP [57]. These findings provide a mechanistic foundation for the current results, where intact autoregulation and preserved compliance correspond to low RAP, while impaired autoregulation coincides with elevated RAP, and exhausted compliance (negative RAP) reflects breakdown of vascular reactivity.

Changes in PbtO_2_ can similarly be explained by this theoretical framework. As RAP transitioned from the impaired to the exhausted state, CBF decreased, resulting in reduced brain tissue oxygenation; thus, PbtO_2_ levels were also lower compared to the RAP > 0.4 state. These findings were mostly consistent across the graphical representation, the analysis of MMM physiology compared to RAP thresholds, and the reverse analysis of RAP against different parameter thresholds. Both the median values and the percentage time spent across different threshold ranges align with the findings from the graphical illustrations. The observed fall in PbtO_2_ underscores that RAP deterioration may serve as an early surrogate for impending brain tissue hypoxia, thereby guiding timely interventions such as optimization of CPP or oxygen delivery strategies.

Secondly, the analysis of the percentage of time spent in different RAP threshold ranges revealed that, across all cases, the majority of the population remained in the RAP > 0.4 state for most of their recording period, as illustrated in Appendix A, suggesting that most TBI patients experienced a high positive RAP, indicating a potential association between TBI and impaired cerebral compliance. The analysis of MMM physiology compared to RAP thresholds also revealed that TBI patients spent a significant amount of time with impaired cerebrovascular reactivity, as a substantial portion of time was associated with positive CA index values. Lastly, formal comparison tests showed that ICP, AMP, MAP, PRx and RAC were significantly different across the RAP states. In the reverse analysis, notably, RAP was significantly different across the thresholds of PRx and RAC. The consistent involvement of these two ICP-derived CA indices in both analyses suggests a potential bidirectional relationship between cerebral autoregulation and cerebral compliance. This mutual association supports the concept that cerebral compliance and autoregulation are interlinked physiological processes, and disturbances in one may reflect or contribute to dysfunction in the other. This finding explains the frequent association between impaired compliance and impaired autoregulation in the majority of TBI patients in this study.

Section 3.3 also reinforces the rationale for the predefined RAP thresholds and established thresholds for other physiological parameters. Appendix A illustrates the percentage time spent by different RAP states within literature-defined parameter thresholds. For instance, considering the threshold PRx at 0, the percentage of time spent in RAP < 0, 0 ≤ RAP ≤ 0.4, and RAP > 0.4 states was 3.12%, 7.69%, and 88.26% when PRx < 0, compared with 6.12%, 12.86%, and 80.86% when PRx > 0. These findings indicate that during compromised autoregulation (PRx > 0), the time spent in RAP < 0 and 0 ≤ RAP ≤ 0.4 states increased, while the time in RAP > 0.4 decreased. This supports the interpretation of RAP < 0 as representing exhausted compliance, 0 ≤ RAP ≤ 0.4 as a transitional state, and PRx > 0 as “compromised” autoregulation. In addition, Appendix A further summarizes the percentage of time spent across different RAP categories relative to thresholds of other parameters.

Lastly, the associations identified between RAP, PbtO_2_, and rSO_2_ are one of the novel findings of this study. To date, few studies have explicitly explored RAP in relation to PbtO_2_, while no prior work has reported a direct association between RAP and rSO_2_ [7,11]. As observed in the previous literature [58,59], a reduction in PbtO_2_ in transitional and exhausted RAP states was also observed in this study. However, prior work has more frequently examined cerebral autoregulation indices in relation to rSO_2_ and PbtO_2_. For rSO_2_, limited studies have demonstrated that impaired autoregulation is associated with reductions in cerebral oximetry. These observations are consistent with the interpretation that NIRS-derived indices serve as surrogates for pulsatile cerebral blood volume (CBV), akin to ICP [60]. Accordingly, the RAP–rSO_2_ association observed here may reflect the well-described RAP–ICP (or RAP–CBV) relationships, linking compensatory reserve to fluctuations in intracranial blood volume. By contrast, PbtO_2_ reflects extracellular oxygen diffusion and cellular utilization. Prior studies have shown that impaired autoregulation is frequently coupled with deteriorating PbtO_2_ levels [58,59]. In this context, the present finding that worsening RAP is associated with reductions in PbtO_2_ suggests that impaired compensatory reserve may compromise cerebral oxygen delivery.

Taken together, these associations provide translational relevance, suggesting that RAP could serve as an integrative marker, not only of compliance but also of oxygen-related physiologic stability. Specifically, impaired RAP states may act as early indicators of downstream derangements in oxygenation (as captured by PbtO_2_ and rSO_2_), thereby supporting the potential role of RAP in real-time bedside monitoring to anticipate oxygen-related secondary insults in TBI management.

### 4.2. Comparative Clustering Patterns of the Parameters

The consistency in clustering across both AHC and PCA suggests stable underlying relationships among physiological signals. When comparing ICP-derived parameters with NIRS-derived parameters, rSO_2_ was the closest to RAP, after ICP and AMP, which may reflect RAP’s sensitivity to cerebral perfusion and oxygenation changes. Furthermore, when PbtO_2_ was considered alongside ICP-derived parameters, it emerged as the closest to RAP, surpassing even ICP and AMP, suggesting a potentially tighter physiological coupling between brain tissue oxygenation and intracranial compliance. None of the CA indices were closer to RAP than ICP, AMP, rSO_2_, or PbtO_2_, except in sub-group analyses, where PRx was found to be the closest to RAP under the RAP < 0 and 0 ≤ RAP ≤ 0.4 conditions, indicating that cerebrovascular reactivity becomes more functionally linked to compliance in compromised autoregulatory states. In this context, the 0 ≤ RAP ≤ 0.4 range is referred to as a “compromised” autoregulatory state, rather than an intact state. As previously discussed, this range may reflect a transitional phase in which RAP decreases from high positive values, marking the shift from impaired to exhausted cerebral compliance.

Additionally, in the majority of the PCA biplots, RAP appeared slightly offset from the core direction of its groups, suggesting that although it was statistically associated with parameters like ICP and AMP, it also captured additional variance not fully explained by these constituent signals alone. This is consistent with the findings from one of our group’s previous studies.

Although AHC and PCA produced highly similar outcomes, KMCA revealed a slightly different clustering pattern. The result of KMCA was the most consistent across all the resolutions and sub-groups. In this analysis, all CA indices clustered with RAP, along with AMP. This contrasts with the findings from AHC and PCA, which indicated that CA indices were not closely associated with RAP. Additionally, KMCA indicated that RAP was more closely associated with AMP than with ICP.

This contrast in results may be because of the fundamental differences in the underlying methodologies of these three models. While PCA and AHC primarily capture linear associations and are sensitive to Euclidean distance or linear variance, they may miss nonlinear or more subtle functional relationships. KMCA, being kernel-based, is designed to capture nonlinear patterns and higher-dimensional associations, which allows it to detect nonlinear coupling or shared variance structures that linear methods overlook. Furthermore, unlike AHC and PCA, KMCA partitions data strictly based on centroid distances, making it more sensitive to the choice of initial cluster centers and the assumption of spherical cluster structure. Importantly, the relatively modest subgroup sample sizes likely exacerbated these sensitivities, amplifying variability in the clustering output. These methodological and sample size limitations together help explain the divergence in KMCA results, rather than indicating true physiological discrepancies.

### 4.3. High-Frequency Temporal Response Patterns

The VARIMA IRF analysis revealed that ICP-ABP-derived parameters were consistently more responsive to changes in RAP across all resolutions, particularly at the 1-min scale, compared to NIRS-derived parameters and PbtO_2_, suggesting a strong physiological coupling between RAP and ICP-ABP-derived measures, likely reflecting their direct mathematical and physiological interdependence. Interestingly, rSO_2__L and rSO_2__R showed comparatively increased responsiveness at lower resolutions, potentially reflecting delayed or cumulative oxygenation dynamics that become more apparent when fine temporal fluctuations are minimized. Sub-group analysis confirmed similar trends in the RAP > 0.4 group, though limited patient numbers in the RAP < 0 and 0 ≤ RAP ≤ 0.4 groups reduce confidence in the results of these two RAP states.

In the context of Granger causality testing, while the analysis revealed limited overall directionality due to the predominance of non-significant cases, some resolution- and parameter-specific patterns did emerge. The finding that ∆X→∆RAP was more frequently significant for ∆PRx, ∆PAx, and ∆RAC at the 1-min resolution suggests that ICP-derived autoregulatory indices might exert predictive influence on RAP during periods of finer temporal granularity, potentially reflecting faster physiological responses. In contrast, the shift at lower resolutions, where parameters like ICP, AMP, MAP, and CPP more often Granger-caused RAP, might point to broader systemic influences becoming more apparent over longer time windows. However, due to the high prevalence of non-significant results, these associations should be interpreted cautiously and might not reflect consistent causal dynamics across all patient cases.

The cross-correlation analysis demonstrated that ∆RAP fluctuates synchronously with its constituent signals (∆ICP, ∆AMP) and ICP-derived CA indices, supporting RAP’s physiological relevance in real-time cerebrovascular monitoring. However, at lower resolutions, ∆MAP and ∆CPP showed stronger associations, suggesting a greater influence of systemic hemodynamics over longer timescales. Analysis at the sub-group level revealed that in impaired states (RAP < 0 and 0 ≤ RAP ≤ 0.4), NIRS-derived parameters (and also CA indices) showed stronger correlations than ICP-derived CA indices, indicating their potential role in compromised autoregulatory conditions. From a clinical standpoint, this stresses the need for high-frequency data capture to appreciate fast compliance-related changes that may be missed at lower sampling rates.

The differences observed between clustering results and high-frequency temporal analyses reflect the distinct purposes and temporal scales of these methods. The clustering methods quantify overall covariance patterns across the dataset, capturing stable, long-term relationships among variables. Conversely, high-frequency temporal analyses assess dynamic, time-dependent interactions, revealing that RAP responds more immediately to changes in ICP. Thus, the two approaches provide complementary insights: clustering methods reflect global co-variance structure, whereas high-frequency analyses capture temporal responsiveness, helping to interpret both the stable and dynamic physiological relationships in TBI patients.

### 4.4. Bilateral Consistency of Cerebral Physiology

Across all analyses, parameters recorded from both the left and right hemispheres (rSO_2_, COx, and COx-a) exhibited similar behavior, and in the clustering results, each pair of corresponding hemispheric parameters consistently grouped together. This suggests that the physiological processes they represent, such as cerebral oxygenation (rSO_2_) and cerebrovascular reactivity (COx and COx-a), are similarly regulated or influenced in both hemispheres, and are likely to exert a comparable influence on RAP and vice versa. This consistency indicates that unilateral monitoring may often suffice for assessing global compliance-related physiology, though bilateral recordings may still be valuable in cases with asymmetric injury patterns.

### 4.5. Impact of Temporal Resolution and RAP Sub-Groups on Analytical Consistency

In both the clustering and high-frequency temporal analyses, lower temporal resolutions were applied in addition to the original 1-min data. Across all models, the clustering results remained largely consistent across resolutions, with only minor variations observed in the AHC and PCA analyses. Similarly, high-frequency response patterns were generally stable across resolutions, with the exception of some differences noted in the Granger causality tests. Overall, these findings suggest that co-variance structures and high-frequency temporal dynamics are largely preserved regardless of the resolution, indicating that these patterns are not strongly influenced by the sampling frequency.

Moreover, the sub-group analysis revealed that deviations from the overall population results were more frequent in the RAP < 0 and 0 ≤ RAP ≤ 0.4 states, whereas the RAP > 0.4 state largely mirrored the full cohort findings. This is likely because RAP < 0 reflects a state of exhausted compliance, and 0 ≤ RAP ≤ 0.4 represents a transitional phase toward this exhaustion. These conditions mark the late stages of cerebrovascular dysfunction. Consequently, physiological relationships in these states may become more unstable and variable across individuals, resulting in greater inconsistency in how cerebral parameters relate to RAP. This variability underlines the clinical challenge of interpreting RAP in late-stage or exhausted compliance, where individualized monitoring and adaptive thresholds may be necessary rather than relying on fixed cut-off values.

### 4.6. Prognostic Significance of Parameter Behaviours in RAP States

The observed patterns in specific RAP states, including RAP < 0 (exhausted compliance) and 0 ≤ RAP ≤ 0.4 (transitional compliance), are mostly consistent with the theoretical progression of cerebral compensatory reserve. For example, in the exhausted compliance state, ICP values were elevated, AMP was reduced, CPP tended to decrease, and CA indices consistently indicated impaired cerebrovascular reactivity. Similarly, PbtO_2_ and NIRS-derived oxygenation measures often showed reduced brain tissue oxygenation or altered responsiveness.

These combined parameter patterns reflect critical depletion of cerebral compliance and impaired autoregulation, which may predict a higher risk of secondary brain injury or poor neurological outcomes. Likewise, in the transitional RAP range (0 ≤ RAP ≤ 0.4), these parameters highlight a phase in which patients are moving from impaired to exhausted compliance. Monitoring these trends during this phase may allow early identification of deterioration, thus providing a potential window for timely interventions.

Therefore, integrating RAP with multimodal monitoring, including ICP, AMP, CPP, CA indices, and oxygenation measures, not only offers physiological insight but also serves as a potential prognostic tool in the management of TBI patients.

## 5. Limitations

Even though this study yielded several important findings, it was not without limitations. Firstly, although the ICP-ABP-derived parameters included 379 data points, the NIRS-derived parameters collectively had only 133, and PbtO_2_ had 116, which are relatively small sample sizes for drawing reliable conclusions from this analysis. Secondly, in the sub-group analysis, the RAP < 0 and 0 ≤ RAP ≤ 0.4 categories often yielded results that differed from those observed in the RAP > 0.4 group and the overall population. These discrepancies may be caused by the fact that the former two states (RAP < 0 and 0 ≤ RAP ≤ 0.4) reflect a condition of exhausted cerebral compliance, where cerebrovascular dysfunction is present, potentially leading to altered physiological relationships. However, these differences could also be attributed to the limited data available for these states. Dividing patients into subgroups inevitably reduced sample sizes, which limited statistical power and restricted the generalizability of the findings, particularly in the smaller RAP < 0 and 0 ≤ RAP ≤ 0.4 groups. Nevertheless, this investigation was conducted on the largest available multi-center high-frequency multimodal monitoring dataset to date, with the next largest being the CENTER-TBI high-resolution ICU cohort, which contains approximately 200 viable patient datasets. Taken together, these findings emphasize both the necessity of careful interpretation of subgroup analyses and the need for larger, multi-center collaborative studies to validate and expand upon these results while preserving adequate statistical power.

Next, RAP, particularly when calculated at higher temporal resolutions, is susceptible to artifacts arising from transient signal disturbances, sensor noise, or abrupt physiological changes. Such artifacts can potentially compound the results by introducing spurious fluctuations in RAP, which may affect both descriptive and temporal analyses. In this study, manual artifact removal was applied by experts in this field, which includes the exclusion of clearly erroneous signal segments and preprocessing to remove extreme outliers using custom scripts. Nevertheless, residual artifacts may still influence high-resolution RAP measurements, and caution should be exercised when interpreting fine-scale temporal dynamics. Future studies could benefit from more advanced artifact detection algorithms and multimodal validation to further ensure the reliability of high-frequency RAP data.

Furthermore, much of this work focuses on the linear population-wide modeling of data. Unique facts surrounding individual patients are lost, with the methodological assumption that sustained biologically meaningful relationships will be consistent and similar across the population. As databases increase with MMM, and a better understanding of the driving factors behind individualized measure responses is gained, work in this area needs to explore selective patient characteristics to more completely document real-time interactions.

Lastly, like all work exploring TBI, there are numerous heterogeneous factors associated with the patient that could impact cerebral pathophysiology. These factors, such as pre-existing neurological conditions, incomplete datasets, or early mortality, can introduce bias and denote underlying patient differences that could impact RAP as a measure. Future research should explore these relationships and document their impact.

## 6. Conclusions

In this analysis, we set out to describe the multivariate relationship between RAP and other aspects of cerebral physiology. There were consistent patterns of the cerebral physiologic parameters across the three predefined RAP states for the majority of cases, which were also observed in the reverse analysis (i.e., consistent pattern of RAP values across the sub-groups of other parameter thresholds). Furthermore, both the clustering and high-frequency temporal response analyses revealed physiologically coherent and largely stable patterns across all examined temporal resolutions. These findings collectively suggest that RAP maintains a consistent and meaningful relationship with other cerebral physiological parameters. Additionally, the observed differences between the impaired and exhausted compliance states highlight the potential of these relationships to inform bedside monitoring, patient trajectory modeling, and future interventional studies. While further research is needed to translate these findings into clinical practice, this analysis demonstrates RAP’s potential as a valuable physiological marker.

## Figures and Tables

**Figure 1 bioengineering-12-01006-f001:**
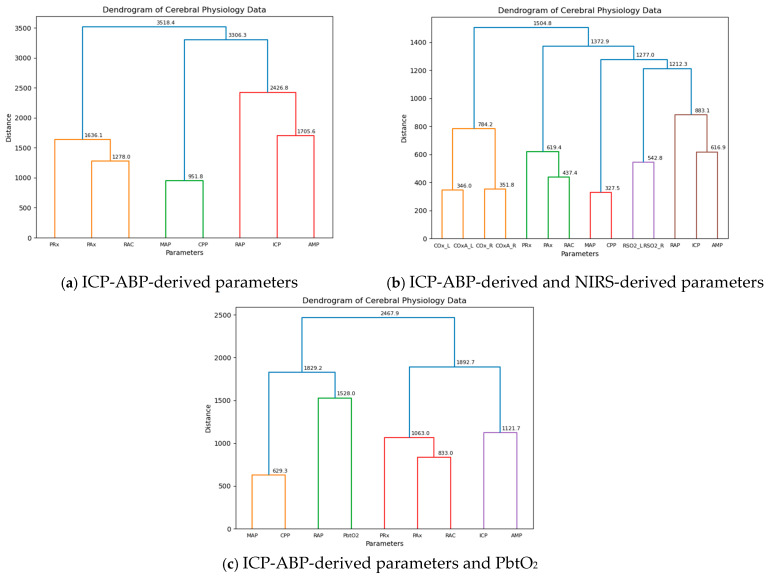
Dendrograms at 1-min resolution across whole population. The figure demonstrates the dendrograms obtained by AHC applications to (**a**) ICP-ABP-derived parameters, (**b**) ICP-ABP-derived and NIRS-derived parameters, and (**c**) ICP-ABP-derived parameters with PbtO_2_. Among all the parameters, ICP, AMP and PbtO_2_ (in (**c**) dendrogram) were in the same clusters with RAP. rSO_2__L and rSO_2__R were in the next closest clusters to RAP(in (**b**) dendrogram). ABP, arterial blood pressure; AMP, pulse amplitude of ICP; COx_L, cerebral oxygenation index of left hemisphere; COx_R, cerebral oxygenation index of right hemisphere; COx-a_L, COx with ABP of left hemisphere; COx-a_R, COx with ABP of the right hemisphere; CPP, cerebral perfusion pressure; ICP, intracranial pressure; MAP, mean arterial pressure; NIRS, near-infrared spectroscopy; PAx, pulse amplitude index; PbtO_2_, brain tissue oxygenation; PRx, pressure reactivity index; RAC, a cerebral autoregulation index; RAP, index of cerebral compensatory reserve; rSO_2__L, regional cerebral oxygen saturation of left hemisphere; rSO_2__R, regional cerebral oxygen saturation of the right hemisphere.

**Figure 2 bioengineering-12-01006-f002:**
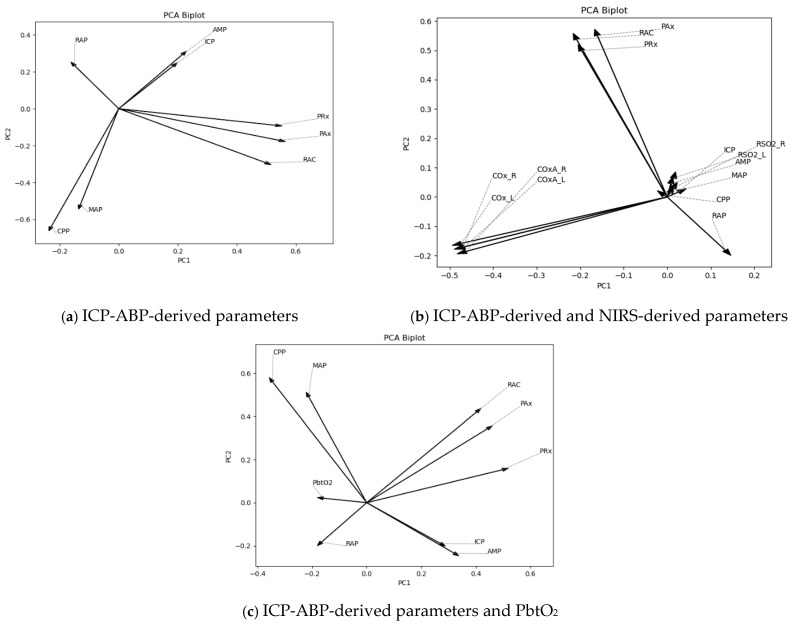
PCA biplots at 1-min resolution across whole population. The figure documents the PCA biplots of (**a**) ICP-ABP-derived parameters, (**b**) ICP-ABP-derived and NIRS-derived parameters, and (**c**) ICP-ABP-derived parameters with PbtO_2_. Among all the parameters, ICP, AMP and PbtO_2_ (in (**c**) biplot) aligned in the same direction as RAP within the PC1–PC2 axis system. rSO_2__L and rSO_2__R were the next closest to RAP in direction (in (**b**) biplot). Similar to findings from dendrograms. ABP, arterial blood pressure; AMP, pulse amplitude of ICP; COx_L, cerebral oxygenation index of left hemisphere; COx_R, cerebral oxygenation index of right hemisphere; COx-a_L, COx with ABP of left hemisphere; COx-a_R, COx with ABP of the right hemisphere; CPP, cerebral perfusion pressure; ICP, intracranial pressure; MAP, mean arterial pressure; NIRS, near-infrared spectroscopy; PAx, pulse amplitude index; PbtO_2_, brain tissue oxygenation; PRx, pressure reactivity index; RAC, a cerebral autoregulation index; RAP, index of cerebral compensatory reserve; rSO_2__L, regional cerebral oxygen saturation of left hemisphere; rSO_2__R, regional cerebral oxygen saturation of the right hemisphere.

**Figure 3 bioengineering-12-01006-f003:**
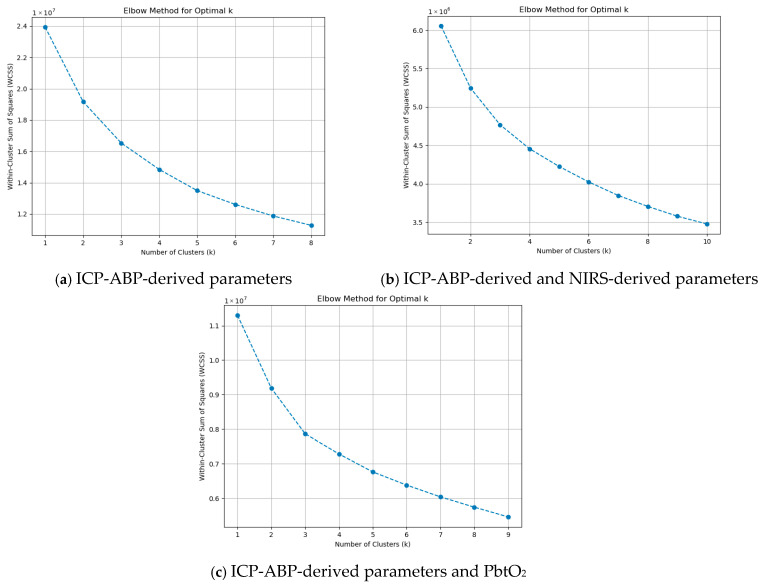
Application of the elbow method at 1-min resolution across whole population. The figure presents the number of clusters vs. WCSS plots of (**a**) ICP-ABP-derived parameters, (**b**) ICP-ABP-derived and NIRS-derived parameters, and (**c**) ICP-ABP-derived parameters with PbtO_2_. All the plots showed, after k = 3, the rate of decrease in WCSS (Within-Cluster Sum of Squares) diminished considerably. ABP, arterial blood pressure; ICP, intracranial pressure; k, number of clusters; NIRS, near-infrared spectroscopy; PbtO_2_, brain tissue oxygenation; WCSS, within-cluster sum of squared errors.

**Table 1 bioengineering-12-01006-t001:** Patient demographics.

Variable	Median (IQR) or Number (%)
Duration of recording (minutes)	5811.23 (2872.90–9951.18)
Number of Patients	379
Age (years)	38 (24–55)
Sex (Male)	295 (77.83%)
GCS	6 (3–7)
GCS Motor	4 (1–5)
**Pupils**	
Bilateral Reactive	250 (65.96%)
Unilateral Unreactive	63 (16.62%)
Bilateral Unreactive	56 (14.77%)
**Marshall CT Classification**	
**VI**	6 (1.58%)
**V**	104 (27.44%)
**IV**	43 (11.34%)
**III**	90 (23.74%)
**II**	118 (31.13%)
**Parameter groups**	
Patient with ICP-ABP-derived parameters	379 (100%)
Patient with combined NIRS-derived parameters	133 (35.09%)
Patient with PbtO_2_	116 (30.60%)
**RAP states**	
RAP < 0	4.78% (2.52–9.72%)
0 ≤ RAP ≤ 0.4	10.91% (5.97–17.19%)
RAP > 0.4	84.31% (72.17–91.29%)

ABP, arterial blood pressure; CT, computerized tomography; GCS, Glasgow Coma Score; ICP, intracranial pressure; IQR, interquartile range; NIRS, near-infrared spectroscopy; PbtO_2_, brain tissue oxygenation; RAP, index of cerebral compensatory reserve.

**Table 2 bioengineering-12-01006-t002:** Slope of the piecewise linear regression.

Parameters	RAP < 0	0 ≤ RAP ≤ 0.4	RAP > 0.4
ICP	−3.43	0.92	3.73
AMP	−0.24	1.09	1.81
MAP	1.85	−1.05	2.25
CPP	5.38	−1.95	−1.5
PRx	−0.08	0.0	−0.59
PAx	0.23	0.21	−0.25
RAC	−0.12	−0.08	−1.09
rSO_2__L	−2.51	0.14	−3.24
rSO_2__R	−1.57	1.44	−2.59
COx_L	0.02	−0.03	−0.21
COx_R	−0.05	0.03	−0.19
COx-a_L	−0.12	−0.02	−0.05
COx-a_R	−0.12	−0.02	−0.03
PbtO_2_	−2.43	−1.22	9.58

AMP, pulse amplitude of ICP; COx_L, cerebral oxygenation index of left hemisphere; COx_R, cerebral oxygenation index of right hemisphere; COx-a_L, COx with ABP of left hemisphere; COx-a_R, COx with ABP of the right hemisphere; CPP, cerebral perfusion pressure; ICP, intracranial pressure; MAP, mean arterial pressure; PAx, pulse amplitude index; PbtO_2_, brain tissue oxygenation; PRx, pressure reactivity index; RAC, a cerebral autoregulation index; RAP, index of cerebral compensatory reserve; rSO_2__L, regional cerebral oxygen saturation of left hemisphere; rSO_2__R, regional cerebral oxygen saturation of the right hemisphere.

**Table 3 bioengineering-12-01006-t003:** The cophenetic correlations for the dendrograms across all the resolutions.

Data Groups	1-min	5-min-by-5-min	10-min-by-10-min	30-min-by-30-min	Hour-by-Hour
ICP-ABP-derived parameters	0.89	0.89	0.89	0.89	0.89
ICP-ABP-derived and NIRS-derived parameters	0.9	0.89	0.89	0.89	0.88
ICP-ABP-derived parameters and PbtO_2_	0.9	0.86	0.85	0.84	0.84

ABP, arterial blood pressure; ICP, intracranial pressure; NIRS, near-infrared spectroscopy; PbtO_2_, brain tissue oxygenation.

**Table 4 bioengineering-12-01006-t004:** Clusters generated from KMCA across all resolutions.

Resolution	ICP-ABP-Derived Parameters	ICP-ABP-Derived and NIRS-Derived Parameters	ICP-ABP-Derived Parameters and PbtO_2_
1-min	0: [‘ICP’], 1: [‘MAP’, ‘CPP’], 2: [‘RAP’, ‘AMP’, ‘PRx’, ‘PAx’, ‘RAC’]	0: [rSO_2__L’, rSO_2__R’], 1: [‘RAP’, ‘ICP’, ‘AMP’, ‘PRx’, ‘PAx’, ‘RAC’, ‘COx_L’, ‘COx_R’, ‘COx-a_L’, ‘COx-a_R’], 2: [‘MAP’, ‘CPP’]	0: [‘ICP’, PbtO_2_], 1: [‘MAP’, ‘CPP’], 2: [‘RAP’, ‘AMP’, ‘PRx’, ‘PAx’, ‘RAC’
5-min-by-5-min	0: [‘ICP’], 1: [‘MAP’, ‘CPP’], 2: [‘RAP’, ‘AMP’, ‘PRx’, ‘PAx’, ‘RAC’]	0: [rSO_2__L’, rSO_2__R’], 1: [‘RAP’, ‘ICP’, ‘AMP’, ‘PRx’, ‘PAx’, ‘RAC’, ‘COx_L’, ‘COx_R’, ‘COx-a_L’, ‘COx-a_R’], 2: [‘MAP’, ‘CPP’]	0: [‘ICP’, PbtO_2_], 1: [‘MAP’, ‘CPP’], 2: [‘RAP’, ‘AMP’, ‘PRx’, ‘PAx’, ‘RAC’]
10-min-by-10-min	0: [‘ICP’], 1: [‘MAP’, ‘CPP’], 2: [‘RAP’, ‘AMP’, ‘PRx’, ‘PAx’, ‘RAC’]	0: [rSO_2__L’, rSO_2__R’], 1: [‘RAP’, ‘ICP’, ‘AMP’, ‘PRx’, ‘PAx’, ‘RAC’, ‘COx_L’, ‘COx_R’, ‘COx-a_L’, ‘COx-a_R’], 2: [‘MAP’, ‘CPP’]	0: [‘ICP’, PbtO_2_], 1: [‘MAP’, ‘CPP’], 2: [‘RAP’, ‘AMP’, ‘PRx’, ‘PAx’, ‘RAC’]
30-min-by-30-min	0: [‘ICP’], 1: [‘MAP’, ‘CPP’], 2: [‘RAP’, ‘AMP’, ‘PRx’, ‘PAx’, ‘RAC’]	0: [rSO_2__L’, rSO_2__R’], 1: [‘RAP’, ‘ICP’, ‘AMP’, ‘PRx’, ‘PAx’, ‘RAC’, ‘COx_L’, ‘COx_R’, ‘COx-a_L’, ‘COx-a_R’], 2: [‘MAP’, ‘CPP’]	0: [‘ICP’, PbtO_2_], 1: [‘MAP’, ‘CPP’], 2: [‘RAP’, ‘AMP’, ‘PRx’, ‘PAx’, ‘RAC’]
Hour-by-hour	0: [‘ICP’], 1: [‘MAP’, ‘CPP’], 2: [‘RAP’, ‘AMP’, ‘PRx’, ‘PAx’, ‘RAC’]	0: [rSO_2__L’, rSO_2__R’], 1: [‘RAP’, ‘ICP’, ‘AMP’, ‘PRx’, ‘PAx’, ‘RAC’, ‘COx_L’, ‘COx_R’, ‘COx-a_L’, ‘COx-a_R’], 2: [‘MAP’, ‘CPP’]	0: [‘ICP’, PbtO_2_], 1: [‘MAP’, ‘CPP’], 2: [‘RAP’, ‘AMP’, ‘PRx’, ‘PAx’, ‘RAC’]

In this table, the clusters were represented by numbers—“0”, “1”, and “2” and the parameters within a cluster were listed inside a bracket. ABP, arterial blood pressure; AMP, pulse amplitude of ICP; COx_L, cerebral oxygenation index of left hemisphere; COx_R, cerebral oxygenation index of right hemisphere; COx-a_L, COx with ABP of left hemisphere; COx-a_R, COx with ABP of the right hemisphere; CPP, cerebral perfusion pressure; ICP, intracranial pressure; MAP, mean arterial pressure; PAx, pulse amplitude index; PbtO_2_, brain tissue oxygenation; PRx, pressure reactivity index; RAC, a cerebral autoregulation index; RAP, index of cerebral compensatory reserve; rSO_2__L, regional cerebral oxygen saturation of left hemisphere; rSO_2__R, regional cerebral oxygen saturation of the right hemisphere.

**Table 5 bioengineering-12-01006-t005:** Responsiveness of RAP to the orthogonal impulse of cerebral physiological parameters.

Direction	1-min		5-min-by-5-min		10-min-by-10-min	
	>0.1%	NA	>0.1%	NA	>0.1%	NA
∆ICP→∆RAP	77.84% (295)	0	63.73% (239)	4	65.50% (243)	8
∆AMP→∆RAP	75.60% (285)	2	62.30% (233)	5	59.84% (222)	8
∆MAP→∆RAP	73.74% (278)	2	68.36% (255)	6	59.14% (220)	7
∆CPP→∆RAP	71.43% (270)	1	67.56% (252)	6	57.99% (214)	10
∆PRx→∆RAP	78.99% (297)	3	53.60% (201)	4	44.09% (164)	7
∆PAx→∆RAP	78.72% (296)	3	52.13% (196)	3	40.32% (150)	7
∆RAC→∆RAP	80.32% (302)	3	56.60% (210)	8	42.32% (157)	8
∆rSO_2__L→∆RAP	72.79% (107)	9	67.65% (92)	20	65.91% (87)	24
∆rSO_2__R→∆RAP	74.31% (107)	11	72.39% (97)	21	61.36% (81)	23
∆COx_L→∆RAP	71.33% (102)	9	42.11% (56)	19	34.59% (46)	19
∆COx_R→∆RAP	74.47% (105)	11	47.37% (63)	19	33.59% (44)	21
∆COx-a_L→∆RAP	77.86% (109)	0	47.69% (62)	10	34.62% (45)	10
∆COx-a_R→∆RAP	77.54% (107)	2	47.29% (61)	11	32.56% (42)	11
∆PbtO_2_→∆RAP	64.60% (73)	3	69.37% (77)	5	67.59% (73)	8

To calculate the percentage values in this table, the NA cases were excluded from the total number of patients. ∆, first-order differenced; AMP, pulse amplitude of ICP; COx_L, cerebral oxygenation index of left hemisphere; COx_R, cerebral oxygenation index of right hemisphere; COx-a_L, COx with ABP of left hemisphere; COx-a_R, COx with ABP of the right hemisphere; CPP, cerebral perfusion pressure; ICP, intracranial pressure; MAP, mean arterial pressure; NA, not applicable; PAx, pulse amplitude index; PbtO_2_, brain tissue oxygenation; PRx, pressure reactivity index; RAC, a cerebral autoregulation index; RAP, index of cerebral compensatory reserve; rSO_2__L, regional cerebral oxygen saturation of left hemisphere; rSO_2__R, regional cerebral oxygen saturation of the right hemisphere.

**Table 6 bioengineering-12-01006-t006:** Granger causality testing of ∆X and ∆RAP pairs across the resolutions for whole population.

Parameters	Direction	1-min	5-min	10-min
∆ICP & ∆RAP	∆ICP→∆RAP	8.97% (34)	40.63% (154)	28.31% (107)
	∆RAP→∆ICP	54.62% (207)	20.05% (76)	15.87% (60)
	NS	36.41% (138)	39.31% (149)	55.82% (211)
	NA	0	0	1
∆AMP & ∆RAP	∆AMP→∆RAP	12.4% (47)	54.09% (205)	32.54% (123)
	∆RAP→∆AMP	44.33% (168)	16.36% (62)	22.22% (84)
	NS	43.27% (164)	29.55% (112)	45.24% (171)
	NA	0	0	1
∆MAP & ∆RAP	∆MAP→∆RAP	14.78% (56)	47.23% (179)	28.57% (108)
	∆RAP→∆MAP	42.22% (160)	13.46% (51)	14.02% (53)
	NS	43.01% (163)	39.31% (149)	57.41% (217)
	NA	0	0	1
∆CPP & ∆RAP	∆CPP→∆RAP	16.09% (61)	46.17% (175)	29.1% (110)
	∆RAP→∆CPP	37.47% (142)	13.72% (52)	17.72% (67)
	NS	46.44% (176)	40.11% (152)	53.17% (201)
	NA	0	0	1
∆PRx & ∆RAP	∆PRx→∆RAP	30.87% (117)	14.78% (56)	12.17% (46)
	∆RAP→∆PRx	27.44% (104)	19.26% (73)	15.34% (58)
	NS	41.69% (158)	65.96% (250)	72.49% (274)
	NA	0	0	1
∆PAx & ∆RAP	∆PAx→∆RAP	33.51% (127)	11.35% (43)	8.99% (34)
	∆RAP→∆PAx	22.69% (86)	24.54% (93)	17.2% (65)
	NS	43.8% (166)	64.12% (243)	73.81% (279)
	NA	0	0	1
∆RAC & ∆RAP	∆RAC→∆RAP	38.52% (146)	12.66% (48)	9.26% (35)
	∆RAP→∆RAC	24.01% (91)	24.27% (92)	22.49% (85)
	NS	37.47% (142)	63.06% (239)	68.25% (258)
	NA	0	0	1
∆rSO_2__L & ∆RAP	∆rSO_2__L→∆RAP	11.35% (16)	23.74% (33)	14.6% (20)
	∆RAP→∆rSO_2__L	45.39% (64)	21.58% (30)	16.06% (22)
	NS	43.26% (61)	54.68% (76)	69.34% (95)
	NA	15	17	19
∆rSO_2__R & ∆RAP	∆rSO_2__R→∆RAP	13.67% (19)	19.57% (27)	11.76% (16)
	∆RAP→∆rSO_2__L	40.29% (56)	20.29% (28)	16.91% (23)
	NS	46.04% (64)	60.14% (83)	71.32% (97)
	NA	16	17	19
∆COx_L & ∆RAP	∆COx_L→∆RAP	10.71% (15)	7.97% (11)	11.03% (15)
	∆RAP→∆COx_L	28.57% (40)	25.36% (35)	13.97% (19)
	NS	60.71% (85)	66.67% (92)	75.0% (102)
	NA	12	14	16
∆COx_R & ∆RAP	∆COx_R→∆RAP	15.11% (21)	5.07% (7)	6.62% (9)
	∆RAP→∆COx_R	20.14% (28)	26.09% (36)	17.65% (24)
	NS	64.75% (90)	68.84% (95)	75.74% (103)
	NA	13	14	16
∆COx-a_L & ∆RAP	∆COx-a_L→∆RAP	11.03% (15)	6.72% (9)	9.85% (13)
	∆RAP→∆COx-a_L	34.56% (47)	20.9% (28)	16.67% (22)
	NS	54.41% (74)	72.39% (97)	73.48% (97)
	NA	4	6	8
∆COx-a_R & ∆RAP	∆COx-a_R→∆RAP	19.26% (26)	8.96% (12)	5.3% (7)
	∆RAP→∆COx-a_R	27.41% (37)	23.13% (31)	17.42% (23)
	NS	53.33% (72)	67.91% (91)	77.27% (102)
	NA	5	6	8
∆PbtO_2_ & ∆RAP	∆PbtO_2_→∆RAP	8.7% (10)	15.93% (18)	16.07% (18)
	∆RAP→∆PbtO_2_	16.52% (19)	16.81% (19)	14.29% (16)
	NS	74.78% (86)	67.26% (76)	69.64% (78)
	NA	1	3	4

To calculate the percentage values in this table, the NA cases were excluded from the total number of patients. ∆, first-order differenced; AMP, pulse amplitude of ICP; COx_L, cerebral oxygenation index of left hemisphere; COx_R, cerebral oxygenation index of right hemisphere; COx-a_L, COx with ABP of left hemisphere; COx-a_R, COx with ABP of the right hemisphere; CPP, cerebral perfusion pressure; ICP, intracranial pressure; MAP, mean arterial pressure; NA, not applicable, NS, not significant; PAx, pulse amplitude index; PbtO_2_, brain tissue oxygenation; PRx, pressure reactivity index; RAC, a cerebral autoregulation index; RAP, index of cerebral compensatory reserve; rSO_2__L, regional cerebral oxygen saturation of left hemisphere; rSO_2__R, regional cerebral oxygen saturation of the right hemisphere.

**Table 7 bioengineering-12-01006-t007:** ∆RAP-∆X cross-correlation analysis across the resolutions for whole population.

Parameter Pair	1-min			5-min			10-min		
	Maximum Correlation	Maximum lag	NA	Maximum Correlation	Maximum Lag	NA	Maximum Correlation	Maximum Lag	NA
∆RAP-∆ICP	0.086 (0.062–0.122)	1.0 (0.0–5.0)	0	0.128 (0.095–0.179)	1.0 (0.0–4.0)	0	0.165 (0.118–0.237)	1.0 (0.0–4.0)	0
∆RAP-∆AMP	0.115 (0.081–0.166)	0.0 (0.0–4.0)	0	0.165 (0.109–0.222)	0.0 (0.0–1.0)	0	0.205 (0.14–0.289)	0.0 (0.0–3.0)	0
∆RAP-∆MAP	0.062 (0.044–0.1)	2.0 (0.0–6.0)	0	0.157 (0.1–0.238)	0.0 (0.0–3.0)	0	0.199 (0.139–0.315)	0.0 (0.0–6.0)	0
∆RAP-∆CPP	0.064 (0.044–0.1)	2.0 (0.0–6.0)	0	0.142 (0.095–0.227)	1.0 (0.0–4.0)	0	0.18 (0.124–0.289)	0.0 (0.0–6.5)	0
∆RAP-∆PRx	0.143 (0.075–0.236)	0.0 (0.0–0.0)	0	0.191 (0.116–0.3)	0.0 (0.0–3.0)	0	0.218 (0.148–0.334)	0.0 (0.0–6.0)	0
∆RAP-∆PAx	0.152 (0.084–0.244)	0.0 (0.0–0.0)	0	0.167 (0.1–0.264)	0.0 (0.0–4.0)	0	0.198 (0.128–0.288)	0.0 (0.0–6.5)	0
∆RAP-∆RAC	0.194 (0.11–0.306)	0.0 (0.0–0.0)	0	0.264 (0.158–0.376)	0.0 (0.0–0.0)	0	0.301 (0.188–0.432)	0.0 (0.0–1.0)	0
∆RAP-∆rSO_2__L	0.079 (0.053–0.121)	3.0 (1.0–8.0)	9	0.15 (0.105–0.22)	2.0 (0.0–12.0)	9	0.179 (0.124–0.253)	3.0 (0.5–10.0)	9
∆RAP-∆rSO_2__R	0.075 (0.05–0.132)	3.0 (1.0–7.0)	10	0.132 (0.104–0.208)	4.0 (0.0–14.0)	10	0.185 (0.126–0.264)	4.0 (0.0–11.0)	10
∆RAP-∆COx_L	0.073 (0.052–0.108)	4.0 (1.0–11.75)	6	0.126 (0.095–0.179)	4.0 (1.0–12.0)	6	0.178 (0.131–0.267)	5.0 (0.0–12.0)	6
∆RAP-∆COx_R	0.064 (0.043–0.098)	4.0 (1.0–10.0)	7	0.127 (0.093–0.193)	6.0 (1.0–15.0)	7	0.175 (0.123–0.268)	5.0 (1.0–12.0)	7
∆RAP-∆COx-a_L	0.074 (0.052–0.114)	2.0 (0.0–9.0)	4	0.135 (0.103–0.181)	3.0 (0.0–8.0)	4	0.175 (0.128–0.265)	4.0 (0.0–12.0)	4
∆RAP-∆COx-a_R	0.065 (0.044–0.122)	3.0 (1.0–8.0)	5	0.129 (0.09–0.21)	5.0 (0.0–12.0)	5	0.169 (0.128–0.269)	6.0 (1.0–12.0)	5
∆RAP-∆PbtO_2_	0.086 (0.062–0.122)	6.0 (2.0–12.0)	1	0.107 (0.076–0.149)	5.0 (1.0–12.0)	1	0.137 (0.099–0.201)	4.0 (1.0–12.0)	2

To calculate the percentage values in this table, the NA cases were excluded from the total number of patients. ∆, first-order differenced; AMP, pulse amplitude of ICP; COx_L, cerebral oxygenation index of left hemisphere; COx_R, cerebral oxygenation index of right hemisphere; COx-a_L, COx with ABP of left hemisphere; COx-a_R, COx with ABP of the right hemisphere; CPP, cerebral perfusion pressure; ICP, intracranial pressure; MAP, mean arterial pressure; NA, not applicable; PAx, pulse amplitude index; PbtO_2_, brain tissue oxygenation; PRx, pressure reactivity index; RAC, a cerebral autoregulation index; RAP, index of cerebral compensatory reserve; rSO_2__L, regional cerebral oxygen saturation of left hemisphere; rSO_2__R, regional cerebral oxygen saturation of the right hemisphere.

## Data Availability

The original contributions presented in this study are included in the article/Appendix A. Further inquiries can be directed to the corresponding author.

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
