# Peer review of "Relationship Between RAP and Multi-Modal Cerebral Physiological Dynamics in Moderate/Severe Acute Traumatic Neural Injury: A CAHR-TBI Multivariate Analysis"

_bioengineering, 2025, doi:10.3390/bioengineering12091006_

Round 1

Reviewer 1 Report

Comments and Suggestions for Authors

Dear Author,

It is an interesting work; however, it is too lengthy and may make difficult for the readers to get engaged.

  • Please ensure the references to appendix items are consistent and clear. For example, the same term “Figure 1” is used for figures in different appendices without specifying the appendix (e.g., “Figure 1 of Appendix A” vs. “Figure 1”), which may lead to confusion.
  • Please use a standardized referencing format: such as “Figure A1”, “Table B2”, “Figure C3”, etc.
  • Please check that all the references are up to date and as per journal style.
  • The results section although presents substantial data, it needs to be expanded for detailed immediate interpretation or clinical contextualization.
  • Although, potentially significant findings are described but need addition/explanation of their clinical implications to ensure to directly engage the reader with the significance of the findings and each major result implies physiologically or methodologically, not just reporting the statistical trends.
  • Please describe the importance of discrepancies or outliers (e.g., unexpected ICP or CPP values in certain RAP states), the likely cause and possible implications. For example: “This increase in ICP under RAP < 0 conditions suggests potential deterioration in compensatory reserve, which may have prognostic significance in TBI management.”
  • The repeated phrases i.e. “discussed later” or “explained below”; may be concisely interpreted in context, and can be expanded later in the Discussion if necessary.
  • Please check for grammar and language flow like
  • Terminology and formatting (e.g., “minute-by-minute,” “semi-supervised,” etc.), recheck the use of passive voice to enhance readability and engagement.
Comments on the Quality of English Language

Need minor editing

Author Response

Reviewer 1

Comment 1: Please ensure the references to appendix items are consistent and clear. For example, the same term “Figure 1” is used for figures in different appendices without specifying the appendix (e.g., “Figure 1 of Appendix A” vs. “Figure 1”), which may lead to confusion.

Please use a standardized referencing format: such as “Figure A1”, “Table B2”, “Figure C3”, etc.

Response 1: Thank you for highlighting this issue.

All the numbering and references of Tables and Figures of the appendices were changed according to the standardized referencing format. For example, in the first page of Appendix A, “Figure A Scatterplots across the entire population with piecewise linear regression (based on RAP thresholds) for the physiological variables at minute-by-minute resolution” was changed to “Figure A.1 Scatterplots across the entire population with piecewise linear regression (based on RAP thresholds) for the physiological variables at minute-by-minute resolution

 and in the manuscript at 494-495 lines in section 3.3, “Afterwards, the median values of different physiological parameters, along with IQR were calculated across the three RAP states and presented in Table A.1 of Appendix A” to “Afterwards, the median values of different physiological parameters, along with IQR were calculated across the three RAP states and presented in Table A of Appendix A.” In the manuscript, these changes were applied in sections 3.2, 3.3, 3.4, 3.5 and 4.1.

Comment 2: Please check that all the references are up to date and as per journal style.

Response 2: Thank you, we have since updated to MDPI format and ensured all references are up to date.

Comment 2: The results section, although it presents substantial data, needs to be expanded for detailed immediate interpretation or clinical contextualization.

Although, potentially significant findings are described but need addition/explanation of their clinical implications to ensure to directly engage the reader with the significance of the findings and each major result implies physiologically or methodologically, not just reporting the statistical trends.

Response 2: Thank you for your valuable feedback regarding this issue.  

  We have restructured the Results and Discussion sections to hopefully provide a better clinical interpretation of the results as well as improve interpretation. To this end we have made substantial changes to the results and discussion section to highlight the following. Lines 473-484 in section 3.2 describe the trends and statistical differences in CA indices across RAP ranges:

Theoretically, as cerebral compliance worsens, CA indices should increase and re-main positive, indicating impaired cerebrovascular reactivity…………… This was due to the transitional phase of cerebral compliance when it worsens from an impaired to an exhausted state.

and 504-507 in section 3.3 stating:

 “Furthermore, consistent with previous figures, the median values of all CA indices indicated that impaired autoregulation (i.e., higher CA index values) was associated with lower RAP values, with the exception of PAx.”

The clinical interpretation of these findings is then elaborated in the Discussion at lines 826–839 in section 4.1, stating

In the context of CA indices, theoretical expectations indicate that worsening compliance is associated with positive CA indices…………… Therefore, the analyses showed that CA indices were higher when RAP was near zero or negative, compared to when RAP was greater than 0.4.

Llines 733–740), the Results present the statistical outputs:

The table presents the percentage and count of cases classified as belonging to the “more responsive” group for each parameter pair across resolutions. ……… The only notable exception was rSO₂_L and rSO₂_R, which demonstrated comparatively higher responsiveness percentages at the lower resolutions.” The corresponding clinical interpretation is then provided in section 4.3 (lines 854–861), where we explain that “The VARIMA IRF analysis revealed that ICP-ABP-derived parameters were consistently more responsive to changes in RAP across all resolutions, particularly at the minute-by-minute scale, ………. potentially reflecting delayed or cumulative oxygenation dynamics that become more apparent when fine temporal fluctuations are minimized.

In a similar manner to the two examples provided, the clinical interpretations of all other important and significant findings are presented in the Discussion section. The Results section is limited to reporting the numerical and statistical outputs only. Including clinical explanations within the Results would result in redundancy.

However, in the Discussion section, some additional explanations were included for key findings to enable readers to engage more directly with their (i.e., the findings) physiological or methodological implications. They are as follows:

At lines 821-824 in section 4.1:

Clinically, this suggests that RAP can serve as an integrated marker reflecting the combined impact of raised ICP, reduced AMP, and diminished CPP, thereby providing clinicians with a more comprehensive indicator of cerebral reserve status rather than considering each parameter in isolation.

At lines 840-843 in section 4.1:

“This transitional behaviour also highlights the importance of not interpreting RAP thresholds in isolation but rather in the broader temporal and physiological trajectory of the patient, as misclassification may lead to delayed recognition of compliance exhaustion.”

At lines 850-852 in section 4.1:

“The observed fall in PbtO₂ underscores a clinically actionable implication: RAP deterioration may serve as an early surrogate for impending brain tissue hypoxia, thereby guiding timely interventions such as optimization of CPP or oxygen delivery strategies.”

At lines 933-935 in section 4.3:

“From a clinical standpoint, this stresses the need for high-frequency data capture to appreciate fast compliance-related changes that may be missed at lower sampling rates.”

At lines 953-955 in section 4.4:

“This consistency indicates that unilateral monitoring may often suffice for assessing global compliance-related physiology, though bilateral recordings may still be valuable in cases with asymmetric injury patterns.”

At lines 973-974 in section 4.5:

“This variability underlines the clinical challenge of interpreting RAP in late-stage or exhausted compliance, where individualized monitoring and adaptive thresholds may be necessary rather than relying on fixed cut-off values”

Comment 3: Please describe the importance of discrepancies or outliers (e.g., unexpected ICP or CPP values in certain RAP states), the likely cause and possible implications. For example: “This increase in ICP under RAP < 0 conditions suggests potential deterioration in compensatory reserve, which may have prognostic significance in TBI management.”

Response 3: Thank you for pointing this out.

Indeed, there were few discrepancies in certain RAP states, particularly RAP < 0 (exhausted compliance) and transitional 0 ≤ RAP ≤ 0.4 states. But the particular discrepancy mentioned by the reviewer, which is,“This increase in ICP under RAP < 0 conditions suggests potential deterioration in compensatory reserve,” is not a discrepancy. Theoretically, it is expected that ICP would rise further, eventually cross the critical threshold value in the exhausted compliance state.

To help highlight this somewhat conceptually confusing factor we have added lines 829 -845:

The conducted analyses aligned with these expectations in the exhausted RAP state but not in the RAP > 0.4 and 0 ≤ RAP ≤ 0.4 ranges. This discrepancy can be explained by the typical progression of TBI patients into an exhausted compliance state. Initially, in an impaired compliance phase, patients may experience further deterioration, leading to arteriolar dilation failure and discontinuity in cerebral blood flow (CBF). This reduces the transmission of arterial pulse pressure to the intracranial compartment, resulting in decreased AMP. Consequently, RAP begins to decline and may pass through the 0 ≤ RAP ≤ 0.4 range, which is normally indicative of intact compliance, but in this context, it represents a transitional phase toward negative RAP values, indicating exhausted compliance. Therefore, the analyses showed that CA indices were higher when RAP was near zero or negative, compared to when RAP was greater than 0.4. This transitional behaviour also highlights the importance of not interpreting RAP thresholds in isolation but rather in the broader temporal and physiological trajectory of the patient, as misclassification may lead to delayed recognition of compliance exhaustion.

However, by discrepancy or outlier, the reviewer may have been referring to the consistent patterns displayed by certain parameters in the exhausted compliance state, as these patterns could hold prognostic value in TBI management, in that context following section was added to the manuscript to address the issue at lines 976-992 in section 4.6:

4.6 Prognostic Significance of Parameter Behaviours in RAP States

The observed patterns in specific RAP states, including RAP < 0 (exhausted compliance) and 0 ≤ RAP ≤ 0.4 (transitional compliance), are mostly consistent with the theoretical progression of cerebral compensatory reserve. For example, in the exhausted compliance state, ICP values were elevated, AMP was reduced, CPP tended to decrease, and CA indices consistently indicated impaired cerebrovascular reactivity. Similarly, PbtO₂ and NIRS-derived oxygenation measures often showed reduced brain tissue oxygenation or altered responsiveness.

These combined parameter patterns reflect critical depletion of cerebral compliance and impaired autoregulation, which may predict higher risk of secondary brain injury or poor neurological outcomes. Likewise, in the transitional RAP range (0 ≤ RAP ≤ 0.4), these parameters highlight a phase in which patients are moving from impaired to exhausted compliance. Monitoring these trends during this phase may allow early identification of deterioration, providing a potential window for timely interventions.

Therefore, integrating RAP with multimodal monitoring, including ICP, AMP, CPP, CA indices, and oxygenation measures, not only offers physiological insight but also serves as a potential prognostic tool in the management of TBI patients.”

Comment 4: The repeated phrases i.e. “discussed later” or “explained below”; may be concisely interpreted in context, and can be expanded later in the Discussion if necessary.

Response 4: Thank you for this thoughtful comment.

Such an instance was found at section 3.2:

“A more detailed explanation regarding this is given in section 4.1.”

However, prior to that, the meaning or implication of the result was already briefly explained by the following, at lines 481-483 in section 3.2:

“These findings suggested that impaired reactivity was associated with lower values of RAP. This was due to the transitional phase of cerebral compliance when it worsens from an impaired to an exhausted state.”

Therefore, only the line “A more detailed explanation regarding this is given in section 4.1.” was removed to address this comment.

Comment 5: Please check for grammar and language flow like

Terminology and formatting (e.g., “minute-by-minute,” “semi-supervised,” etc.), recheck the use of passive voice to enhance readability and engagement.

Response 5: In terms of terminology, all of them were rechecked. They were consistent throughout the manuscript.

To enhance the readability and engagement, the following sentences were changed to passive voice:

At lines 808-809 in section 4

“First, we performed a detailed descriptive analysis to assess statistical associations across different sub-groups and threshold ranges.”

to “First, a detailed descriptive analysis to assess statistical associations across different sub-groups and threshold ranges was performed.”

At lines 809-810 in section 4

“Next, we applied semi-supervised machine learning techniques to examine parameter clustering, with a particular focus on how RAP grouped with other variables.”

to “Next, clustering techniques were applied to examine parameter clustering, with a particular focus on how RAP grouped with other variables.”

At lines 811-812 in section 4

“Finally, we analyzed high-frequency temporal response patterns to evaluate how RAP dynamically responds to changes in other MMM parameters. Several key aspects deserve highlighting.”

to “Finally, high-frequency temporal response patterns were analyzed to evaluate how RAP dynamically responds to changes in other MMM parameters. Several key aspects deserve highlighting.”

Finally, we changed minute-by-minute to 1-minute, and semi-supervised was removed to only cases where it has a clear contextual meaning with common methodologies.

Reviewer 2 Report

Comments and Suggestions for Authors

Abstract: The abstract introduces a range of complex statistical and machine learning techniques (PCA, VARIMA, Granger causality) without briefly stating their purpose or outcomes. The link between impaired RAP and elevated intracranial pressure or reduced pulse amplitude, but they are not presented with sufficient effect sizes, p-values. The conclusion introduces the concept of patient trajectory modeling without describing what this entails in practical terms or how RAP would be integrated into such approaches.

Introduction lacks a clear hypothesis. Additionally, while the introduction mentions machine learning (ML) methods (semi-supervised clustering), it does not explain why these are suitable for the data.  There is some redundancy in explaining the progression from intact to exhausted compliance, which could be tightened for conciseness. A clearer structure required separating the physiological background, limitations of current approaches, and objectives of this specific study.

Methods lacks detailed information about the nature and resolution of the "high-frequency physiological data" collected. Specifying sampling rates, duration of monitoring, and preprocessing steps (filtering or artifact removal).

The inclusion criteria are appropriate and consistent with prior TBI research, but there is no mention of exclusion criteria for example, patients with pre-existing neurological conditions, incomplete datasets, or early mortality, which could introduce bias. The mention of pupillary reactivity and CT classification is useful, but how these factors were used in the analysis (as covariates or descriptive) is not specified. Briefly mention data security protocols, how de-identification was ensured, especially given the multi-institutional data sharing.

The Discussion and Conclusion sections of this study present an in-depth multivariate analysis of cerebral compliance (RAP) and its relationship with other multimodal monitoring (MMM) parameters in TBI patients.The discussion effectively synthesizes how RAP tracks with real-time changes in intracranial physiology, identifying its potential role as a surrogate marker for deteriorating cerebral compliance and autoregulation. In the Discussion, while emphasizing the translational importance of anatomical localization for functional and physiological monitoring tools in brain injury or disease recent studies should be discussed in this manner. The present study examining the multivariate behavior of the RAP index in TBI patients supports the growing need for integrative approaches that combine structural and physiological brain data. This aligns modeling approach where accurate spatial mapping can inform the interpretation of physiological changes in neurocritical conditions and when discussing anatomical variations or extracranial-intracranial vascular communications influencing ICP/RAP from: Mathematical and Dynamic Modeling of the Anatomical Localization of the Insula in the Brain. Neuroinform 23, 29 (2025). The conclusion aligns well with this by reinforcing RAP’s translational potential for bedside monitoring and prognostic modeling. However, practical clinical implications and actionable thresholds for RAP usage are not clearly defined, and the call for “future interventional studies” remains general.Despite some analytical complexity and dataset limitations, the manuscript is a valuable contribution and supports RAP’s utility in neurocritical care.Missing references should be added and checked in the text.

Comments on the Quality of English Language

 The English could be improved to more clearly express the research

Author Response

Reviewer 2

Comment 1: Abstract: The abstract introduces a range of complex statistical and machine learning techniques (PCA, VARIMA, Granger causality) without briefly stating their purpose or outcomes.

Response 1: Thank you for identifying this.

For AHC, PCA, and KMCA, their application was justified as a means to explore multivariate covariance patterns, as noted at lines 45-47 in the Abstract

“Agglomerative hierarchical clustering, principal component analysis, and kernel-based clustering were applied to explore multivariate covariance structures.”

In contrast, the rationale for applying VARIMA IRF analysis, Granger causality testing, and cross-correlation analysis was not specified. Accordingly, the following change was made at lines 47-50 in Abstract

From “Also, vector autoregressive integrated moving average impulse response functions, cross-correlation, and Granger causality.”

To “Then, high-frequency temporal analyses, including vector autoregressive integrated moving average impulse response functions (VARIMA IRF), cross-correlation, and Granger causality, were performed to assess dynamic coupling between RAP and other physiological signals.”

Moreover, the outcomes derived from these analyses were also not reported in the Abstract. To address this, the following statements were added at lines 54-56

“Clustering analyses consistently grouped RAP with its constituent signals (ICP and AMP), followed by brain oxygenation parameters (brain tissue oxygenation (PbtO₂) and regional cerebral oxygen saturation (rSO₂)).”

and at lines 58-64

“High-frequency temporal analyses revealed that RAP showed comparatively stronger responsiveness to ICP- and arterial blood pressure (ABP)-derived parameters at minute-by-minute resolution. Moreover, in comparison between ICP-derived and near-infrared spectroscopy (NIRS)-derived CA indices, the former were closer to RAP in clustering, and RAP demonstrated greater sensitivity to changes in these ICP-derived CA indices in high-frequency temporal analyses. These trends remained consistent at lower temporal resolutions as well.”

Comment 2: The link between impaired RAP and elevated intracranial pressure or reduced pulse amplitude, but they are not presented with sufficient effect sizes, p-values.

Response 2: Thank you for highlighting this issue.

To address this, the following change was applied at lines 50-53 in Abstract

From “Impaired and exhausted RAP state was associated with elevated intracranial pressure, and exhausted associated with reduced pulse amplitude.”

To “Impaired and exhausted RAP state was associated with elevated intracranial pressure (p = 0.021). In terms of AMP, impaired RAP was associated with elevated AMP, but exhausted RAP was associated with reduced pulse amplitude (p = 3.94e-09).”

Comment 3: The conclusion introduces the concept of patient trajectory modeling without describing what this entails in practical terms or how RAP would be integrated into such approaches.

Response 3: Thank you for your thoughtful comment. To resolve this concern, the following change was applied at lines 66-72 in Abstract

From “These findings support RAP’s potential as a valuable metric for bedside monitoring and its prospective role in guiding patient trajectory modeling and interventional studies in TBI.”

To “Integrating RAP into patient trajectory modelling and developing predictive frameworks based on these findings across different RAP states can map the evolution of cerebral physiology over time, which may improve prognostication and guide individualized interventions in TBI management. Therefore, these findings support RAP’s potential as a valuable metric for bedside monitoring and its prospective role in guiding patient trajectory modeling and interventional studies in TBI.”

Comment 4: Introduction lacks a clear hypothesis.

Response 4: Thank you for your thoughtful comment.

To present a clear hypothesis, the following was added at lines 126-135 in the introduction section:

“If analyzed, impaired RAP (i.e., positive RAP values) will be associated with increased ICP and AMP. Cerebral autoregulation (CA) measurements will be more positive in impaired RAP states. Furthermore, worsening RAP values (i.e., more positive values) will be associated with decreased cerebral perfusion pressure (CPP), mean arterial pressure (MAP), regional cerebral oxygen saturation (rSO2), and brain tissue oxygenation (PbtO2). While these relationships are likely to be more apparent in higher resolution data, similar trends should still be evident in lower-resolution datasets. It’s because even when high-frequency fluctuations are averaged out in lower-resolution data, the underlying patterns, such as shifts in physiologies associated with impaired compliance, remain preserved.”

Comment 5: Additionally, while the introduction mentions machine learning (ML) methods (semi-supervised clustering), it does not explain why these are suitable for the data. 

Response 5: Thank you for identifying this.

In the introduction, only the types of analyses to be conducted are outlined, such as clustering methods for quantifying covariance patterns in multivariate space and high-frequency temporal response assessments. However, the specific algorithms employed for each analysis are not detailed there, as these are described in the Materials and Methods section.

Thus, AHC, PCA, and KMCA were first introduced in Section 2.7, where their selection was also justified. To provide a broader perspective, the following addition was made at lines 305–308 in Section 2.7:

“These three methods allow both dimensionality reduction and identification of subgroups within the data, thereby enabling a more robust characterization of shared variance structures and covariance patterns across parameters. Therefore, these three algorithms were chosen for this section.”

Comment 6: There is some redundancy in explaining the progression from intact to exhausted compliance, which could be tightened for conciseness.

Response 6: Thank you for pointing this out.

To eliminate redundancy, the section was revised accordingly, and the updated version now reads as follows at lines 91-99:

“The RAP index ranges from -1 to +1. For RAP , as ICP rises, an intact cerebrovascular reactivity mechanism prevents significant changes in AMP; therefore, RAP remains close to 0. However, when ICP continues to increase and cerebrovascular reactivity becomes impaired, AMP also begins to rise (correlating to ICP),leading to higher positive RAP values, which indicates compromised compliance. If ICP keeps increasing and exceeds a critical threshold, cerebral vessels reach a maximum vasodilation state and cerebrovascular function breaks down, leading to reduced transmission of pulse pressure, i.e., reduced AMP. This leads to lower RAP value (less correlated AMP and ICP) and is the characteristic of exhausted cerebral compliance.”

Comment 7: A clearer structure required separating the physiological background, limitations of current approaches, and objectives of this specific study.

Response 7: Thank you we have adjusted the introduction, to hopefully better address this comment, Noting that the physiologic background of RAP is represented by the following at lines 87-99:

“One such physiological marker, the cerebral compliance or compensatory reserve index (RAP), has emerged as a promising indicator of cerebral compensatory reserve (and therefore compliance). RAP is derived as the moving Pearson correlation coefficient between ICP and its fundamental pulse amplitude (AMP). Here fundamental component refers to the first dominant frequency peak of ICP pulse in frequency domain. The RAP index ranges from -1 to +1. In the intact state, as ICP rises, an intact cerebrovascular reactivity mechanism prevents significant changes in AMP; therefore, RAP remains close to 0. However, when ICP continues to increase and cerebrovascular reactivity becomes impaired, AMP also begins to rise, leading to a high positive RAP value, which indicates compromised compliance. If ICP keeps increasing and exceeds a critical threshold, cerebral vessels reach a maximum vasodilation state and cerebrovascular function breaks down, leading to reduced transmission of pulse pressure, i.e., reduced AMP. This leads to a negative RAP value, the characteristic of exhausted cerebral compliance.”

The limitations of current approaches are outlined in the following section (lines 106–120), with a few additional sentences added to provide further clarity and context:

“Despite its potential, the clinical adoption of RAP in TBI cases has been limited, largely due to an incomplete understanding of its characteristics. Previous inves-tigations have focused on characterizing the temporal dynamics of RAP and its con-stituent signals, ICP and AMP, as well as addressing artifact management strategies.19 However, there remains a notable gap in the literature regarding the broader cerebral physiology context and the injury burden of impaired RAP in association with other continuous multi-modal monitoring (MMM) cerebral physiology.19 A recent systematic review highlighted that various MMM physiological indices consistently exhibited dis-tinct patterns across different states of compliance (i.e., intact, impaired, exhausted) in TBI patients, emphasizing the potential for a more granular analysis of how RAP aligns with these physiologic parameters.11 However, none of the prior studies had RAP as their primary focus. Furthermore, some of the studies had a small dataset or excluded too many subjects, and therefore, their results were not significant enough to apply the findings clinically. Additionally, a few of the studies showed inconsistent findings that did not match the majority of the studies.”

The objectives and hypothesis of this specific study are outlined at lines 121-137:

“Therefore, this study aims to characterize the insult burden of impaired cerebral compensatory reserve in moderate/severe TBI patients in relation to other critical aspects of cerebral physiology by A. describing general descriptive associations between RAP and other physiologic signals, B. exploring multivariate covariance patterns using clus-tering techniques, and C. assessing high-resolution RAP responses to dynamic changes in related variables. If analyzed, impaired RAP (i.e., positive RAP values) will be asso-ciated with increased ICP and AMP. Cerebral autoregulation (CA) measurements will be more positive in impaired RAP states. Furthermore, worsening RAP values (i.e., more positive values) will be associated with decreased cerebral perfusion pressure (CPP), mean arterial pressure (MAP), regional cerebral oxygen saturation (rSO2), and brain tissue oxygenation (PbtO2). While these relationships are likely to be more apparent in higher-resolution data, similar trends should still be evident in lower-resolution da-tasets. It’s because even when high-frequency fluctuations are averaged out in low-er-resolution data, the underlying patterns, such as shifts in physiologies associated with impaired compliance, remain preserved. The broader objective is to understand the overall burden of impaired RAP in the context of cerebral physiology, thereby laying the groundwork for its future use in bedside monitoring, patient trajectory modeling, and interventional clinical studies.”

Comment 8: Methods lacks detailed information about the nature and resolution of the "high-frequency physiological data" collected. Specifying sampling rates, duration of monitoring, and preprocessing steps (filtering or artifact removal).

Response 8: Thank you for highlighting this.

The nature of the data for different physiological signals is mentioned in sections 2.3 (Physiologic Data Acquisition) and 2.4 (Signal Processing)5.

Several signals were derived from ICP and ABP. Their sampling rate was added at lines 182-183 in section 2.3:

“As an exemplar in Manitoba, within the ICU setting, ICP and ABP signals were sampled at 100 Hz from analog outputs.”

To clarify the resolution of the data, the following was added at line 220 in section 2.4:

The primary derived data was at minute-by-minute intervals.

To mention the duration of monitoring, the following was added at lines 202-203 in section 2.4;

“Median data monitoring duration for each patient was 5811.23s with an IQR of (2872.90s—9951.18s)”

Preprocessing steps, like filtering, artifact removal, are mentioned in section 2.5 (Data cleaning).

Comment 9: The inclusion criteria are appropriate and consistent with prior TBI research, but there is no mention of exclusion criteria for example, patients with pre-existing neurological conditions, incomplete datasets, or early mortality, which could introduce bias.

Response 9: Thank you for this comment and although we agree that as the pathophysiological understanding surrounding high frequency MMM grows; analysis and work subsectioning the data into more unique categories will be important. However, given that we are still exploring the foundational aspects of RAP and the limited number of patients in this study; we felt that so long as the patient met the inclusion criteria, they would be included in this study. Future research should explore the exclusion criteria; thus we have added the following to limitations:

Line 1023 –

Lastly, like all work exploring TBI there are numerous heterogeneous factors associated with the patient that could impact cerebral pathophysiology. Factors like pre-existing neurological conditions, incomplete datasets, or early mortality, which could introduce bias and denote underlying patient differences that could impact RAP as a measure. Future research should explore these relationships and document their impact.

Comment 10: The mention of pupillary reactivity and CT classification is useful, but how these factors were used in the analysis (as covariates or descriptive) is not specified.

Response 10: Thank you for highlighting this.

Pupillary reactivity and CT classification were recorded for all patients as part of the baseline neurological assessment. These variables were summarized descriptively to provide context regarding patient severity across RAP states. They were not included as covariates in the multivariate or high-frequency temporal analyses, as the primary focus of this study was on the physiological relationships among RAP, ICP, AMP, CPP, and other multimodal monitoring parameters. Future more as past comment indicates, we did not do a sub analysis of the data.

Comment 11: Briefly mention data security protocols, how de-identification was ensured, especially given the multi-institutional data sharing.

Response 11: Thankyou we have added the following:

LINES 145-150 For the data collection, each patient was assigned an anonymized number by the collecting institution (with this patient information being stored only locally in double secured format, as per institutional requirements). The anonymized data is then sent to a single site for storage and compilation (University of Manitoba). For more information we refer the interested reader to the following manuscript.[20] 

Comment 12: The Discussion and Conclusion sections of this study present an in-depth multivariate analysis of cerebral compliance (RAP) and its relationship with other multimodal monitoring (MMM) parameters in TBI patients. The discussion effectively synthesizes how RAP tracks with real-time changes in intracranial physiology, identifying its potential role as a surrogate marker for deteriorating cerebral compliance and autoregulation. In the Discussion, while emphasizing the translational importance of anatomical localization for functional and physiological monitoring tools in brain injury or disease recent studies should be discussed in this manner. The present study examining the multivariate behavior of the RAP index in TBI patients supports the growing need for integrative approaches that combine structural and physiological brain data. This aligns modeling approach where accurate spatial mapping can inform the interpretation of physiological changes in neurocritical conditions and when discussing anatomical variations or extracranial-intracranial vascular communications influencing ICP/RAP from: Mathematical and Dynamic Modeling of the Anatomical Localization of the Insula in the Brain. Neuroinform 23, 29 (2025). The conclusion aligns well with this by reinforcing RAP’s translational potential for bedside monitoring and prognostic modeling. However, practical clinical implications and actionable thresholds for RAP usage are not clearly defined, and the call for “future interventional studies” remains general. Despite some analytical complexity and dataset limitations, the manuscript is a valuable contribution and supports RAP’s utility in neurocritical care. Missing references should be added and checked in the text.

Response 12: Thank you for your thoughts and comments.

Reviewer 3 Report

Comments and Suggestions for Authors

RE: Relationship Between RAP and Multi-Modal Cerebral Physiological Dynamics in Moderate/Severe Acute Traumatic Neural Injury: A CAHR-TBI Multivariate Analysis

Summary: The authors evaluated multivariate covariance structures to characterize the potential relationship between cerebral compliance index (RAP) and physiological parameters in moderate-severe traumatic brain injury (TBI) patients. Three clustering methods were applied to assess the association between RAP and cerebral physiological parameters cross three RAP threshold categories of TBI participants. In addition, the autoregressive integrated moving average (ARIMA) model, cross-correlation and Granger causality” were employed to outline the time-series structure of RAP across different temporal resolutions. The results of the relationship between RAP, AMP, ICP were not consistent, i.e. among different clustering methods, VARIMA Impulse Response Function Analysis and Granger causality. Furthermore, although the ideas of applying multivariate analysis is appreciated, it is unclear how useful the results of this study in term of monitoring TBI patients in intensive care would be.

Major:

1. In the abstract, “Also, vector autoregressive integrated moving average impulse response functions, cross-correlation, and Granger causality.” is not a complete sentence.

2. “..reverse analysis was carried out using the thresholds of other physiological parameters, examining how RAP changed in response. Similarly, this was analyzed using the median measurements and statistical comparisons. Furthermore, the percentage time spent of RAP within different RAP states at these threshold ranges was examined. The thresholds for various parameters were as follows: ICP (20 mmHg, 22 mmHg), CPP (60 mmHg, 70 mmHg), rSO2 (60, 70, 80, 90),PbtO2 (15 mmHg, 20 mmHg), PRx (0, +0.25, +0.35), PAx (0, +0.25), RAC (0), COx/COx-a (0, +0.20). “,

It is unclear to me, why there were more than two cut-off values for thresholding in these parameters?

2. “...this involved normalizing each impulse response relative to its original variable. A response was considered “more responsive” if its absolute value exceeded 0.001 (i.e., a ≥0.1% change in the normalized scale) within steps 11 to 15, consistent with thresholds used in prior studies.” What are the ”steps” exactly? “The observation in these steps was effective since immediate responses might relect noise or autoregulatory transients, while sustained changes after a short delay are more likely to reflect true, biologically meaningful interactions. Additionally, observing steps after 15 could reduce the practical utility of the analysis for real-time monitoring.”

3. The term of “machine learning” should be reserved for the analyses including the components of parameters selection, training and cross-validation. Therefore, the “machine learning” in the description of parameter clustering should not be used, e.g. “2.7. Application of Semi-Supervised Machine Learning Methods for Parameter Clustering “ “To quantify covariance patterns in multivariate space for the discussed physiological parameters, semi-supervised machine learning (ML) methods”.

4. Normally in an ARIMA model, we make use of either the AR or MA. It is uncommon to use both ARMA.

5. Possible RAP artifacts, particularly in higher temporal resolution that may compound the results need to be addressed.

5. The authors need to discuss the discrepancy of the findings regarding the association in-between RAP, AMP and ICP. For example, “KMCA indicated that RAP was more closely associated with AMP than with ICP.”, however the results of (VARIMA) Impulse Response Function (IRF) Analysis in Table 5, and the results of Granger causality in Table 6 suggests RAP was more associated with ICP than with AMP.

Author Response

Reviewer 3

Comment 1: In the abstract, “Also, vector autoregressive integrated moving average impulse response functions, cross-correlation, and Granger causality.” is not a complete sentence.

Response 1: Thank you for identifying this.

In light of this concern, and in response to a reviewer’s comment, the following change was made at lines 47-50 in the Abstract.

From “Also, vector autoregressive integrated moving average impulse response functions, cross-correlation, and Granger causality.”

To “Then, high-frequency temporal analyses, including vector autoregressive integrated moving average impulse response functions (VARIMA IRF), cross-correlation, and Granger causality, were performed to assess dynamic coupling between RAP and other physiological signals.”

Comment 2. “..reverse analysis was carried out using the thresholds of other physiological parameters, examining how RAP changed in response. Similarly, this was analyzed using the median measurements and statistical comparisons. Furthermore, the percentage time spent of RAP within different RAP states at these threshold ranges was examined. The thresholds for various parameters were as follows: ICP (20 mmHg, 22 mmHg), CPP (60 mmHg, 70 mmHg), rSO2 (60, 70, 80, 90),PbtO2 (15 mmHg, 20 mmHg), PRx (0, +0.25, +0.35), PAx (0, +0.25), RAC (0), COx/COx-a (0, +0.20). “,

It is unclear to me, why there were more than two cut-off values for thresholding in these parameters?

Response 2: Thank you for the thoughtful comment.

The use of multiple thresholds for the physiological parameters is taken from prior established literature. For example, according to the Brain Trauma Foundation (BTF) guidelines, treating ICP > 22 mmHg is recommended because values above this level are associated with increased mortality.1,2 Additionally, mention of ICP > 20 mmHg was also present in decompressive craniectomy.1,2 In case of CPP, it was recommended that CPP should be maintained between 60 – 70 mmHg.1

In case of the CA indices, the thresholds commonly reported, such as PRx (0, +0.25, +0.35), PAx (0, +0.25), RAC (0), and COx/COx-a (0, +0.20), are not arbitrary cutoffs but rather values that have been examined in prior clinical and experimental studies.2,3 These thresholds emerged from investigations exploring the association between impaired cerebrovascular reactivity and clinically relevant outcomes, including mortality, functional recovery, and secondary injury progression in TBI. For example, positive PRx values above 0 have consistently been interpreted as a loss of autoregulation, with higher thresholds such as +0.25 or +0.35 linked to progressively worse 6-month outcomes and increased risk of physiologic derangements.2,3 Similar reasoning has been applied to PAx, RAC, and COx,4 where studies have empirically identified cutoff points at which impaired autoregulation is most strongly associated with adverse outcomes or loss of slow-wave homeostasis.2,3 Thus, the selected thresholds reflect empirically validated demarcations from previous research, where index values crossing these boundaries corresponded to clinically significant impairment in cerebral autoregulation.

The same is applicable to rSO24,5 and PbtO26–8 as well.

In the manuscript, the following was stated at lines 260-263 in section 2.6, addressing this issue:

“Threshold values for these parameters were determined based on commonly referenced values from previous studies in related fields.”

Which was revised as follows to improve clarity:

“The thresholds used for physiological parameters, including ICP, CPP, CA indices (PRx, PAx, RAC, COx/COx-a), rSO₂, and PbtO₂, were based on prior published studies and established clinical guidelines. These values are not arbitrary but represent empirically validated cutoffs linked to clinically relevant outcomes, such as impaired cerebrovascular reactivity, secondary injury progression, and mortality in TBI patients.

Comment 3. “...this involved normalizing each impulse response relative to its original variable. A response was considered “more responsive” if its absolute value exceeded 0.001 (i.e., a ≥0.1% change in the normalized scale) within steps 11 to 15, consistent with thresholds used in prior studies.” What are the ”steps” exactly? “The observation in these steps was effective since immediate responses might relect noise or autoregulatory transients, while sustained changes after a short delay are more likely to reflect true, biologically meaningful interactions. Additionally, observing steps after 15 could reduce the practical utility of the analysis for real-time monitoring.”

Response 3: Thank you for raising this question.

Each step represents one unit of the model’s temporal resolution (either 1, 5 or 10 minutes depending on the model). In case of resolution, step 1 indicates the immediate response in the first step following the impulse (i.e., 1 minute for 1-minute data), step 2 reflects the response  (i.e., 2 minutes for 1-minute data), and so on. Evaluating steps 11–15, therefore, corresponds to examining the sustained response 11 to 15 minutes after the initial perturbation.

To address this issue, the following sentence was added at lines 392-393 in section 2.8.1:

“To be noted, each step represents one unit of the model’s temporal resolution.”

For this IRF analysis to which all of this is referring, we are interested in exploring that statistically impact that one variable (mainly RAP) has on the other variables. The methodology behind IRF is to model the impact that one variable has to a single pulsed input of another using VARIMA relationship (ie. How a one moment increase in one variable will impact another). Given that we are using overall VARIMA models your concern with reflect noise or autoregulatory transients should not impact the overall model (in theory), and thus like you say the more meaningful sustained biological are noted. Combining with the bootstrapping method, the IRF should in theory documents the meaning full sustained impacts. This however does take into account many assumptions of which we describe in the limitations section as follows:

Lines 1016-1022 Furthermore, much of this work focuses on the linear population wide modeling of data. Unique facts surrounding individual patients are lost, with the methodological assumption that sustained biological meaningful relationship will be consistent and similar across the population, As databases increase with MMM, and better understanding around the driving factors behind individualized measure responses, work in this area needs to explore selective patient characteristics to more completely document live time interations.

Comment 4. The term of “machine learning” should be reserved for the analyses including the components of parameters selection, training and cross-validation. Therefore, the “machine learning” in the description of parameter clustering should not be used, e.g. “2.7. Application of Semi-Supervised Machine Learning Methods for Parameter Clustering “ “To quantify covariance patterns in multivariate space for the discussed physiological parameters, semi-supervised machine learning (ML) methods”.

Response 4: Thank you for pointing this out.

To address this issue, the following changes have been applied throughout the manuscript:

At line 124 in section 1, from “B. exploring multivariate covariance patterns using semi-supervised machine learning (ML) techniques”

To “B. exploring multivariate covariance patterns using clustering techniques”

At line 267 in section 2.7, from “Application of Semi-Supervised Machine Learning Methods for Parameter Clustering”

To “Application of Algorithms for Parameter Clustering”

At line 268-270 in section 2.7, from “To quantify covariance patterns in multivariate space for the discussed physiological parameters, semi-supervised machine learning (ML) methods, namely agglomerative hierarchical clustering (AHC), principal component analysis (PCA), and K-means clustering (KMCA), were utilized.”

To “To quantify covariance patterns in multivariate space for the discussed physiological parameters, clustering methods, namely agglomerative hierarchical clustering (AHC), principal component analysis (PCA), and K-means clustering (KMCA), were utilized.”

At lines 809-810 in section 4, from “Next, semi-supervised machine learning techniques were applied to examine parameter clustering, with a particular focus on how RAP grouped with other variables.”

To “Next, clustering techniques were applied to examine parameter clustering, with a particular focus on how RAP grouped with other variables.”

Comment 5. Normally in an ARIMA model, we make use of either the AR or MA. It is uncommon to use both ARMA.

Response 5: Thank you for your valuable feedback regarding this issue.

While it is true that in many traditional applications of ARIMA modeling, either AR or MA components alone may adequately capture the autocorrelation structure, in physiological time series, including RAP, ICP, and AMP, the data often exhibit both short-term moving average–like effects (due to transient fluctuations or measurement noise) and longer-term autoregressive dependencies (reflecting underlying cerebral physiology and autoregulatory dynamics). Therefore, allowing the model to select both AR and MA terms (ARMA structure) is appropriate when optimizing fit, as it enables the model to capture the full temporal complexity of the signals. The use of both AR and MA terms has also been reported in prior biomedical applications of ARIMA.9,10

The following sentence at lines 351-352 in section 2.8.1 referenced the prior studies that included VARIMA (i.e., vector ARIMA):

“Impulse Response Function (IRF) analysis was then applied to the fitted VARIMA models to quantify the effect of a one-time impulse in a given physiological variable (e.g., ICP, MAP) on RAP over subsequent time points, allowing for the temporal propagation of influence to be assessed.”

Comment 6. Possible RAP artifacts, particularly in higher temporal resolution that may compound the results need to be addressed.

Response 6: Thank you for highlighting this issue.

In our study, artifact management procedures, such as exclusion of erroneous segments and outlier removal, were applied to mitigate these effects. Residual artifacts may still exist, and we have clarified this limitation in the Discussion. To address this issue, the following section was added at lines 1005-1015 in the Limitations section:

“Lastly, RAP, particularly when calculated at higher temporal resolutions, is susceptible to artifacts arising from transient signal disturbances, sensor noise, or abrupt physiological changes. Such artifacts can potentially compound the results by introducing spurious fluctuations in RAP, which may affect both descriptive and temporal analyses. In this study, manual artifact removal was applied by experts in this field, which includes the exclusion of clearly erroneous signal segments and also pre-processing to remove extreme outliers by custom scripts. Nevertheless, residual artifacts may still influence high-resolution RAP measurements, and caution should be exercised when interpreting fine-scale temporal dynamics. Future studies could benefit from more advanced artifact detection algorithms and multimodal validation to further ensure the reliability of high-frequency RAP data.”

Comment 7. The authors need to discuss the discrepancy of the findings regarding the association in-between RAP, AMP and ICP. For example, “KMCA indicated that RAP was more closely associated with AMP than with ICP.”, however the results of (VARIMA) Impulse Response Function (IRF) Analysis in Table 5, and the results of Granger causality in Table 6 suggests RAP was more associated with ICP than with AMP.

Response 7: Thank you for your thoughtful comment regarding this issue.

The apparent discrepancy between KMCA results and the high-frequency temporal analyses (VARIMA IRF and Granger causality) reflects differences in the methodological purpose and temporal resolution of these approaches. KMCA is a clustering algorithm applied to quantify covariance patterns in multivariate analyses over the entire dataset, capturing stable, long-term relationships among variables such as RAP, AMP, and ICP. In contrast, VARIMA IRF and Granger causality are high-frequency temporal analyses designed to evaluate dynamic, time-dependent interactions between signals comparatively at higher resolution. Consequently, while KMCA suggested that RAP clustered more closely with AMP, the high-frequency temporal analyses revealed that RAP responds more immediately to changes in ICP. These findings are not contradictory but complementary: KMCA captures overall co-variance structure, whereas VARIMA IRF and Granger causality capture temporal causality and responsiveness. We have clarified this distinction in the Discussion to prevent misinterpretation.

To address this issue following section was added at lines 936-943 in section 4.3:

“The differences observed between clustering results and high-frequency temporal analyses reflect the distinct purposes and temporal scales of these methods. The clustering methods quantify overall covariance patterns across the dataset, capturing stable, long-term relationships among variables. Conversely, high-frequency temporal analyses assess dynamic, time-dependent interactions, revealing that RAP responds more immediately to changes in ICP. Thus, the two approaches provide complementary insights: clustering methods reflect global co-variance structure, whereas high-frequency analyses capture temporal responsiveness, helping to interpret both the stable and dynamic physiological relationships in TBI patients.”

Round 2

Reviewer 2 Report

Comments and Suggestions for Authors

While the authors emphasize the mutual association between compliance and autoregulation, the mechanistic explanation remains superficial and lacks integration with existing pathophysiological models in traumatic brain injury (TBI). The study highlights clustering results from AHC, PCA, and KMCA, yet the discussion does not sufficiently address why these methods yield divergent outcomes or how methodological limitations (sensitivity to linear vs. nonlinear relationships, sample size effects) may bias interpretations. Additionally, the reliance on subgroup analyses for conditions such as RAP < 0 and 0 ≤ RAP ≤ 0.4 is problematic, as the small sample sizes in these groups significantly weaken the statistical power and undermine the generalizability of conclusions. The terminology of “compromised” autoregulatory states is introduced but not clearly justified or validated against established thresholds in the literature. Moreover, while the authors suggest novel associations between RAP, PbtO₂, and rSO₂, these findings are not compared with broader studies, and no discussion is provided on potential clinical implications or translational value in TBI management. Some references have been cited excessively, including both relevant and irrelevant sources, such as Czosnyka, M., and Zeiler et al.

Comments on the Quality of English Language

The English could be improved to more clearly express the research.

Author Response

Comment1: While the authors emphasize the mutual association between compliance and autoregulation, the mechanistic explanation remains superficial and lacks integration with existing pathophysiological models in traumatic brain injury (TBI).

Thank you for highlighting this issue. To address this, the following is added at lines 844-856 in section 4.1:

“This relationship between cerebral compliance and cerebral autoregulation is also supported by pre-clinical and clinical literature that has examined the pathophysiological basis of ICP–AMP dynamics and their link to autoregulatory function. Experimental studies from the Cambridge group and subsequent animal model investigations in rabbits have demonstrated that as ICP rises toward critical thresholds, a distinct breakpoint in AMP occurs. This inflection point has been theorized to represent the critical closing pressure of small- to medium-sized cerebral vessels. Beyond this point, progressive ICP elevations lead to diminished transmission of arterial pulsatility into the intracranial compartment, reflected by falling AMP values and corresponding negative RAP. These findings provide a mechanistic foundation for the current results, where intact autoregulation and preserved compliance correspond to low RAP, while impaired autoregulation coincides with elevated RAP, and exhausted compliance (negative RAP) reflects breakdown of vascular reactivity.”

Comment2: The study highlights clustering results from AHC, PCA, and KMCA, yet the discussion does not sufficiently address why these methods yield divergent outcomes or how methodological limitations (sensitivity to linear vs. nonlinear relationships, sample size effects) may bias interpretations.

Thank you for identifying this. While AHC and PCA produced comparable clustering outcomes, KMCA demonstrated divergent results. To address this, the following was already included at line 945 in section 4.2:

“This may be because of the fundamental differences in the underlying methodologies of these three models. While PCA and AHC primarily capture linear associations and are sensitive to Euclidean distance or linear variance, therefore, they may miss nonlinear or more subtle functional relationships. KMCA, being kernel-based, is designed to capture nonlinear patterns and higher-dimensional associations, which allows it to detect nonlinear coupling or shared variance structures that linear methods overlook.”

However, to provide a clearer explanation and additional insights, the following was added:

“Furthermore, unlike AHC and PCA, KMCA partitions data strictly based on centroid distances, making it more sensitive to the choice of initial cluster centers and the assumption of spherical cluster structure. Importantly, the relatively modest subgroup sample sizes likely exacerbated these sensitivities, amplifying variability in the clustering output. These methodological and sample size limitations together help explain the divergence in KMCA results, rather than indicating true physiological discrepancies.”

Comment3: Additionally, the reliance on subgroup analyses for conditions such as RAP < 0 and 0 ≤ RAP ≤ 0.4 is problematic, as the small sample sizes in these groups significantly weaken the statistical power and undermine the generalizability of conclusions.

Thank you for pointing this out. Indeed, the data sample size for RAP < 0 and 0 ≤ RAP ≤ 0.4 groups were small, which was already addressed at line 1055 in the Limitations section:

“However, these differences could also be attributed to the limited data available for these states. Acquiring more data in these ranges would help resolve this uncertainty.”

However, to emphasize this limitation and to clearly represent it to the reader, the following was added:

“However, these differences could also be attributed to the limited data available for these states. Dividing patients into subgroups inevitably reduced sample sizes, which limited statistical power and restricted the generalizability of the findings, particularly in the smaller RAP < 0 and 0 ≤ RAP ≤ 0.4 groups. Nevertheless, this investigation was conducted on the largest available multi-center high-frequency multimodal monitoring dataset to date, with the next largest being the CENTER-TBI high-resolution ICU cohort, which contains approximately 200 viable patient datasets. Taken together, these findings emphasize both the necessity of careful interpretation of subgroup analyses and the need for larger, multi-center collaborative studies to validate and expand upon these results while preserving adequate statistical power.”

Comment4: The terminology of “compromised” autoregulatory states is introduced but not clearly justified or validated against established thresholds in the literature.

Thank you for highlighting this. To justify the terminology of “compromised” autoregulatory states, the following was added at lines 884-895 in section 4.1:

“Section 3.3 also reinforces the rationale for the predefined RAP thresholds and established thresholds for other physiological parameters. Table A.2 illustrates the percentage time spent by different RAP states within literature-defined parameter thresholds. For instance, considering the threshold PRx at 0, the percentage of time spent in RAP < 0, 0 ≤ RAP ≤ 0.4, and RAP > 0.4 states was 3.12%, 7.69%, and 88.26% when PRx < 0, compared with 6.12%, 12.86%, and 80.86% when PRx > 0. These findings indicate that during compromised autoregulation (PRx > 0), the time spent in RAP < 0 and 0 ≤ RAP ≤ 0.4 states increased, while the time in RAP > 0.4 decreased. This supports the interpretation of RAP < 0 as representing exhausted compliance and 0 ≤ RAP ≤ 0.4 as a transitional state, while PRx > 0 validates the designation of “compromised” autoregulatory states. In addition, Table A.1 further summarizes the percentage of time spent across different RAP categories relative to thresholds of other parameters.”

Comment5: Moreover, while the authors suggest novel associations between RAP, PbtO₂, and rSO₂, these findings are not compared with broader studies, and no discussion is provided on potential clinical implications or translational value in TBI management.

Thank you for pointing this out. To address this issue, the following was added at lines 896-917 in section 4.1:

“Lastly, the associations identified between RAP, PbtO₂, and rSO₂ is one of the novel findings of this study. To date, few studies have explicitly explored RAP in relation to PbtO₂, while no prior work has reported a direct association between RAP and rSO₂. As observed in the previous literature, a reduction in PbtO₂ in transitional and exhausted RAP states was noticeable in this study. However, prior work has more frequently examined cerebral autoregulation indices in relation to rSO₂ and PbtO₂. For rSO₂, limited studies have demonstrated that impaired autoregulation is associated with reductions in cerebral oximetry. These observations are consistent with the interpretation that NIRS-derived indices serve as surrogates for pulsatile cerebral blood volume (CBV), akin to ICP.[64] Accordingly, the RAP–rSO₂ association observed here may reflect the well-described RAP–ICP (or RAP–CBV) relationships, linking compensatory reserve to fluctuations in intracranial blood volume. By contrast, PbtO₂ reflects extracellular oxygen diffusion and cellular utilization. Prior studies have shown that impaired autoregulation is frequently coupled with deteriorating PbtO₂ levels. In this context, the present finding that worsening RAP is associated with reductions in PbtO₂ suggests that impaired compensatory reserve may compromise cerebral oxygen delivery.

Taken together, these associations provide translational relevance, suggesting that RAP could serve as an integrative marker not only of compliance but also of oxygen-related physiologic stability. Specifically, impaired RAP states may act as early indicators of downstream derangements in oxygenation (as captured by PbtO₂ and rSO₂), thereby supporting the potential role of RAP in real-time bedside monitoring to anticipate oxygen-related secondary insults in TBI management.”

Comment6: Some references have been cited excessively, including both relevant and irrelevant sources, such as Czosnyka, M., and Zeiler et al.

Thank you for pointing this out. Comparatively, irrelevant references have been excluded.
